

# Nonsymmorphic spin-space cubic groups and $SU(2)_1$ conformal invariance in one-dimensional spin-1/2 models

**Wang Yang[1], Alberto Nocera[2], Chao Xu[3,4] and Ian Affleck[2]**

**1** School of Physics, Nankai University, Tianjin, 300071, China
**2** Department of Physics and Astronomy and Stewart Blusson Quantum Matter Institute, University of British Columbia, Vancouver, B.C., Canada, V6T 1Z1
**3** Kavli Institute for Theoretical Sciences, University of Chinese Academy of Sciences, Beijing 100190, China
**4** Institute for Advanced Study, Tsinghua University, Beijing 100084, China

## Abstract

Recently, extended gapless phases with emergent $SU(2)_1$ conformal invariance occupying finite regions in the phase diagrams have been found in one-dimensional spin-1/2 models with nonsymmorphic $O_h$ symmetry groups. In this work, we investigate the question of whether the conditions for emergent $SU(2)_1$ invariance can be loosened. We find that besides the nonsymmorphic $O_h$ group, the other four smaller nonsymmorphic cubic groups including $O$, $T_h$, $T_d$ and $T$ can also give rise to emergent $SU(2)_1$ invariance. Minimal spin-1/2 models having these nonsymmorphic cubic groups as symmetry groups are constructed, and numerical evidences for the emergent $SU(2)_1$ invariance are provided. Our work is useful for understanding gapless phases in one-dimensional spin systems with nonsymmorphic symmetries.

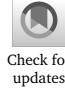
doi:10.21468/SciPostPhys.17.4.097

# 1   Introduction

Nonsymmorphic symmetries are a class of crystalline symmetry operations which involve a combination of fractional lattice translations and rotations or reflections [1]. In recent years, there are increasing research interests in studying the consequences of nonsymmorphic symmetries in condensed matter systems. Among the investigations of nonsymmorphic symmetries, the noninteracting and weakly interacting systems have been well-studied [2–13] including topological insulators, hourglass fermions, Dirac insulators and topological semi-metals, whereas strongly correlated nonsymmorphic systems remain much less explored [14–16]. It is worth to note that there is a special category of nonsymmorphic symmetry groups named "spin-space groups" [17], in which the spins are allowed to rotate independently from the spatial coordinates, different from the usual magnetic space groups where the rotations in the spin and orbital spaces are combined in a spin-orbit coupled manner.

One-dimensional (1D) Kitaev spin models are 1D versions of the generalized Kitaev spin-1/2 models on the honeycomb lattice [18–35], which can be constructed by selecting one or several rows out of the honeycomb lattice. Recent studies on 1D Kitaev spin models (such as Kitaev-Heisenberg-gamma model, Kitaev models with Dzyaloshinskii-Moriya interactions, etc.) have revealed rich nonsymmorphic spin-space symmetry group structures, leading to exotic strongly correlated properties, including emergent conformal symmetries [24,34], non-local string order parameters [22, 28], and magnetic phases with exotic symmetry breaking patterns [24, 25, 28].

Particularly, with the help of a unitary transformation called six-sublattice rotation, the symmetry group of the 1D Kitaev-gamma model [24] has been shown to be isomorphic to the $O_h$ group in the sense of modulo lattice translation symmetries, where $O_h$ is the full octahedral group, the largest crystalline point group with 48 group elements, or more rigorously, the symmetry group $G_{K\Gamma}$ satisfies the non-split short exact sequence $1 \to \mathbb{Z} \to G_{K\Gamma} \to O_h \to 1$ [31]. It has been analytically proved and numerically verified that the nonsymmorphic $O_h$ symmetry stabilizes an extended gapless phase which has an emergent $SU(2)_1$ conformal symmetry at low energies. This is an interesting and exotic result since an extended phase with emergent $SU(2)_1$ conformal symmetry occupying a finite region in the phase diagram usually arises from a full $SU(2)$ symmetry, not discrete symmetry groups. It is worth to note that while the 1D spin-1/2 gamma model lies in the gapless phase, the pure Kitaev model does not [24]. Hence the 1D spin-1/2 gamma model can be viewed as the minimal model realizing the nonsymmorphic $O_h$ symmetry with an emergent $SU(2)_1$ conformal invariance at low energies.

In this work, we investigate the question: Is it possible for a smaller nonsymmorphic symmetry group to stabilize an extended phase of emergent $SU(2)_1$ conformal invariance? We find that the answer to this question is yes, and in fact, the nonsymmorphic counterparts of all the five cubic point groups $O_h$, $O$, $T_h$, $T_d$ and $T$ can lead to emergent $SU(2)_1$ invariance. We note that $T \cong A_4$ is the smallest cubic point group among the five where $A_4$ is the alternating group of order 12. The nonsymmorphic $T$ group is the smallest group which can achieve the goal of stabilizing $SU(2)_1$ invariance, namely, an emergent $SU(2)_1$ invariance is not possible for nonsymmorphic planar groups. For all the five nonsymmorphic cubic groups, minimal models are constructed, which can be viewed variants of the 1D gamma model. Using density matrix renormalization group (DMRG) simulations [36–38], numerical evidence for emergent $SU(2)_1$ invariance are provided for all the minimal models.

It is worth to note that two scenarios need to be distinguished depending on whether the Hamiltonian after the six-sublattice rotation has three-site or six-site periodicities. The minimal model for nonsymmorphic $O_h$ group with a six-site periodicity in the six-sublattice rotated frame can be obtained from the 1D gamma model by adding a Dzyaloshinskii-Moriya interaction [34], from which minimal models of other nonsymmorphic groups with six-site periodicities can be constructed as variants. We find that if the rotated Hamiltonian is three-site periodic, then all the five nonsymmorphic cubic symmetry groups can stabilize an extended $SU(2)_1$ phase. On the other hand, for the six-site periodic case, only the nonsymmorphic $O_h$, $O$, and $T_d$ groups can do the job.

The rest of the paper is organized as follows. In Sec. 2, a brief review is given for the 1D spin-1/2 gamma model and the related nonsymmorphic $O_h$ symmetry, emergent $SU(2)_1$ conformal invariance, and nonsymmorphic nonabelian bosonization formulas. In Sec. 3, the nonsymmorphic $T$ group is constructed, and the existence of an extended gapless phase with an emergent $SU(2)_1$ conformal symmetry is proved. A minimal model realizing the nonsymmorphic $T$ group is also constructed, and DMRG numerical evidence on the $SU(2)_1$ invariance is presented. In Sec. 4, the nonsymmorphic $T_h$ group is constructed, and the corresponding minimal model – the asymmetric gamma model – is discussed in details. Sec. 5 is devoted to discussing the nonsymmorphic $O$ group and the corresponding minimal model. In Sec. 6,

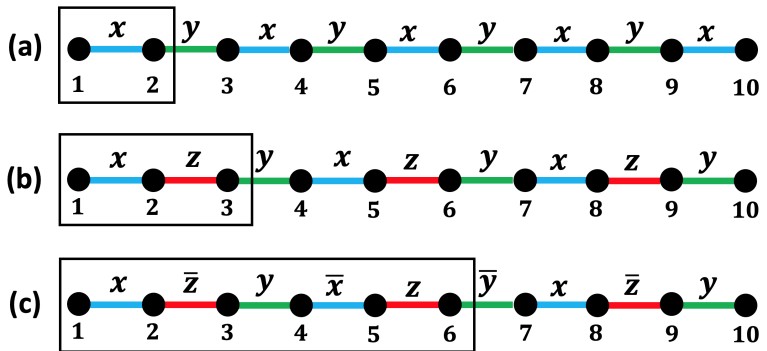

Figure 1: Bond patterns of the Kitaev-gamma chain (a) without sublattice rotation, (b) after the six-sublattice rotation, (c) with a nonzero DM interaction after six-sublattice rotation. The black squares represent the unit cells. This figure is taken from Ref. [34].

the nonsymmorphic $T_d$ symmetry and the corresponding minimal model are constructed and investigated. In Sec. 7, the cases of six-site periodicity are studied, including a review of the $O_h$ case, and investigations of the $O$ and $T_d$ cases. Finally, in Sec. 8, the main results of this work are summarized.

# 2 Review of the 1D symmetric gamma model

In this section, we give a brief review of the 1D spin-1/2 symmetric gamma model, its hidden nonsymmorphic $O_h$ symmetry group structure, and the emergence of SU(2)$_1$ conformal invariance at low energies.

## 2.1 Symmetric gamma model and the nonsymmorphic $O_h$ symmetry

The Hamiltonian of the 1D spin-1/2 symmetric gamma model is defined as

$$H_{S\Gamma} = \sum_{<ij>\in\gamma \text{ bond}} \Gamma\left(S_i^\alpha S_j^\beta + S_i^\beta S_j^\alpha\right),\tag{1}$$

in which $(\alpha, \beta, \gamma)$ form a right-handed coordinate system, and the pattern for the bond $\gamma$ is shown in Fig. 1 (a).

To discuss the symmetry group of the symmetric gamma model, it is useful to consider a unitary transformation $U_6$, called six-sublattice rotation, defined as

$$\text{Sublattice } 1 : (x, y, z) \rightarrow (x', y', z'),$$
$$\text{Sublattice } 2 : (x, y, z) \rightarrow (-x', -z', -y'),$$
$$\text{Sublattice } 3 : (x, y, z) \rightarrow (y', z', x'),$$
$$\text{Sublattice } 4 : (x, y, z) \rightarrow (-y', -x', -z'),$$
$$\text{Sublattice } 5 : (x, y, z) \rightarrow (z', x', y'),$$
$$\text{Sublattice } 6 : (x, y, z) \rightarrow (-z', -y', -x'),\tag{2}$$

in which "Sublattice $i$" ($1 \le i \le 6$) represents all the sites $i + 6n$ ($n \in \mathbb{Z}$) in the chain, and we have abbreviated $S^\alpha$ ($S'^\alpha$) as $\alpha$ ($\alpha'$) for short ($\alpha = x, y, z$). It can be verified that the transformed Hamiltonian $H'_{S\Gamma} = (U_6)^{-1}H_{S\Gamma}U_6$ acquires the following form,

$$H'_{S\Gamma} = \sum_{<ij>\in\gamma \text{ bond}} (-\Gamma)\left(S_i'^\alpha S_j'^\alpha + S_i'^\beta S_j'^\beta\right),\tag{3}$$

in which $(\alpha, \beta, \gamma)$ form a right-handed coordinate system, and the pattern for the bond $\gamma = x, z, y$ is shown in Fig. 1 (b), having a three-site periodicity. Explicit forms of $H_{S\Gamma}$ and $H'_{S\Gamma}$ are included in Appendix A.

In the $U_6$ frame, the Hamiltonian $H'_{S\Gamma}$ is invariant under the following symmetry transformations,

1. $\mathcal{T}$       $: (S_i'^x, S_i'^y, S_i'^z) \to (-S_i'^x, -S_i'^y, -S_i'^z)$,

2. $R_a T_a$       $: (S_i'^x, S_i'^y, S_i'^z) \to (S_{i+1}'^z, S_{i+1}'^x, S_{i+1}'^y)$,

3. $R_I I$       $: (S_i'^x, S_i'^y, S_i'^z) \to (-S_{4-i}'^z, -S_{4-i}'^y, -S_{4-i}'^x)$,

4. $R(\hat{x}, \pi)$    $: (S_i'^x, S_i'^y, S_i'^z) \to (S_i'^x, -S_i'^y, -S_i'^z)$,

5. $R(\hat{y}, \pi)$    $: (S_i'^x, S_i'^y, S_i'^z) \to (-S_i'^x, S_i'^y, -S_i'^z)$,

6. $R(\hat{z}, \pi)$    $: (S_i'^x, S_i'^y, S_i'^z) \to (-S_i'^x, -S_i'^y, S_i'^z)$,

in which $\mathcal{T}$ is the time reversal operation; $I$ is the spatial inversion operation with inversion center located at site 2; $R(\hat{n}, \phi)$ denotes a global spin rotation around $\hat{n}$-direction by an angle $\phi$; $T_a$ denotes the spatial translation by one lattice site; $T_{na}$ represents the translation operator by $n$ sites; $R_a$ is the spin rotation around $(1, 1, 1)$-direction by an angle $-2\pi/3$; and $R_I$ is a $\pi$-rotation around the $(1, 0, -1)$-direction. The symmetry group $G$ is generated by the above symmetry transformations as

$$G_{S\Gamma} = <\mathcal{T}, R_a T_a, R_I I, R(\hat{x}, \pi), R(\hat{y}, \pi), R(\hat{z}, \pi)>, \tag{4}$$

in which $<...>$ represents the group generated by the elements within the bracket. It is worth to note that $G_{S\Gamma}$ is a spin-space group [17], since the rotations are restricted in the spin space, not of a spin-orbit coupled structure (all symmetry groups discussed in later sections in this work are spin-space groups). Since $T_{3a} = (R_a T_a)^3$ generates an abelian normal subgroup of $G_{S\Gamma}$, the quotient $G_{S\Gamma}/<T_{3a}>$ is a group. It has been proved in Ref. [24] that $G_{S\Gamma}$ is a nonsymmorphic group and satisfies

$$G_{S\Gamma}/<T_{3a}> \cong O_h, \tag{5}$$

where $O_h$ is the full octahedral group, which is the largest three-dimensional crystalline point group. The group $G_{S\Gamma}$ satisfies the following short exact sequence,

$$1 \to <T_{3a}> \to G_{S\Gamma} \to O_h \to 1, \tag{6}$$

and the rigorous mathematical meaning of "nonsymmorphic" is that the above short exact sequence is non-split [31].

We note that as discussed in Ref. [26], Eq. (5) has an intuitive understanding by observing that all the symmetry operations in Eq. (4) act as symmetries of a three-dimensional (3D) spin cube when restricted in the spin space as shown in Fig. 2. On the other hand, it is known that the symmetry group of a 3D cube is the $O_h$ group [41], hence it is not a surprise that the symmetry group $G_{S\Gamma}$ is intimately related to the $O_h$ group.

## 2.2 Emergent SU(2)$_1$ conformal invariance

Remarkably, it has been verified by DMRG numerics that $H'_{S\Gamma}$ (equivalently $H_{S\Gamma}$) has an emergent SU(2)$_1$ conformal symmetry at low energies, described by the Sugawara Hamiltonian of the (1+1)-dimensional SU(2)$_1$ Wess-Zumino-Witten (WZW) model,

$$\mathcal{H} = \frac{2\pi}{3} v(\vec{J}_L \cdot \vec{J}_L + \vec{J}_R \cdot \vec{J}_R), \tag{7}$$

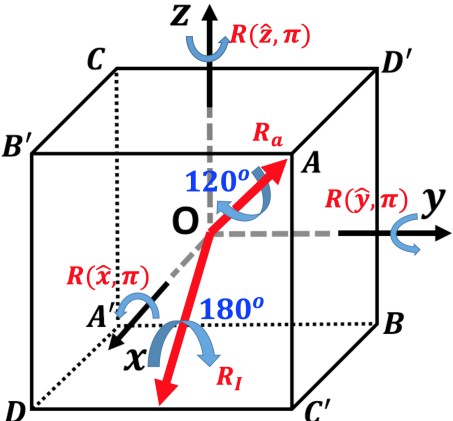

Figure 2: Actions of the symmetry operations in Eq. (4) in the spin space as symmetries of a 3D spin cube. Time reversal operation corresponds to the inversion of the spin cube, which is not shown in the figure. This figure is taken from Ref. [26].

in which $v$ is the velocity; $\vec{J}_L$ and $\vec{J}_R$ are the left and right WZW currents, respectively, defined as

$$\vec{J}_L = -\frac{1}{4\pi}\text{tr}[(\partial_z g)g^\dagger \vec{\sigma}],$$
$$\vec{J}_R = \frac{1}{4\pi}\text{tr}[g^\dagger(\partial_{\bar{z}} g)\vec{\sigma}], \tag{8}$$

where $g$ is the $SU(2)_1$ primary field which is a $2 \times 2$ SU(2) matrix. In addition, the following nonsymmorphic nonabelian bosonization formulas have been proposed in Ref. [24] which build the connections between the spin operators $S_k'^\alpha$ ($k \in \mathbb{Z}$, $\alpha = x, y, z$) and the $SU(2)_1$ WZW degrees of freedom,

$$S_{j+3n}^\alpha = D_{L,j}^\alpha J_L^\alpha + D_{R,j}^\alpha J_R^\alpha + (-)^{j+3n} C_j^\alpha N^\alpha, \tag{9}$$

in which: $N^\alpha = i\text{tr}(g\sigma^\alpha)$ where $\sigma^\alpha$ ($\alpha = x, y, z$) is the Pauli matrix; $n$ is the unit cell index; $1 \le j \le 3$ is the site index within a unit cell; $D_{L,j}^\alpha$, $D_{R,j}^\alpha$, $C_{L,j}^\alpha$, $C_{R,j}^\alpha$ are bosonization coefficients satisfying

$$D_{v,1}^z = D_{v,2}^y = D_{v,3}^x = D_1,$$
$$D_{v,1}^x = D_{v,2}^z = D_{v,3}^y = D_{v,1}^y = D_{v,2}^x = D_{v,3}^z = D_2, \tag{10}$$

and

$$C_{v,1}^z = C_{v,2}^y = C_{v,3}^x = C_1,$$
$$C_{v,1}^x = C_{v,2}^z = C_{v,3}^y = C_{v,1}^y = C_{v,2}^x = C_{v,3}^z = C_2, \tag{11}$$

in which $v = L, R$. In the sense of low energy properties, the above nonsymmorphic nonabelian bosonization formulas apply to any model with nonsymmorphic $O_h$ symmetry and emergent $SU(2)_1$ conformal invariance.

We note that the spin-1/2 symmetric gamma model serves as the minimal model having nonsymmorphic $O_h$ symmetry and emergent $SU(2)_1$ conformal invariance. There are other terms which preserve the nonsymmorphic $O_h$ symmetry and keep the $SU(2)_1$ conformal invariance. An example of such additional terms is the 1D Kitaev term (so that the model becomes the more general Kitaev-gamma model), as discussed in details in Ref. [24].

## 2.3 Generalizations of SU(2)$_1$ invariance to other nonsymmorphic symmetry groups

As previously reviewed, a nonsymmorphic $O_h$ symmetry group leads to an emergent SU(2)$_1$ conformal invariance at low energies. Then a natural question is: Is it possible to lower the symmetries of the symmetric gamma model, while at the same time maintaining the emergent SU(2)$_1$ conformal symmetry? Furthermore, what is the smallest nonsymmorphic symmetry group required to ensure the emergent SU(2)$_1$ conformal invariance? In the following sections, we will answer the above questions, by demonstrating that: 1. the required smallest nonsymmorphic symmetry group is the smallest cubic group, i.e., the $T$ group; 2. all the five cubic groups $O_h$, $O$, $T_h$, $T_d$, $T$ can stabilize the emergent SU(2)$_1$ conformal symmetry.

We note that the symmetric gamma model is the minimal model realizing the $O_h$ nonsymmorphic symmetry. It also serves as the parent model for a number of models, which are minimal models for different nonsymmorphic cubic groups. These other models can be constructed by adding one or several of the following terms to the Hamiltonian $H_{S\Gamma}$,

$$
\begin{aligned}
&\sum_{<ij>\in\gamma\,\text{bond}} D_M(S_i^\alpha S_j^\beta - S_i^\beta S_j^\alpha), \\
&\sum_{<ij>\in\gamma\,\text{bond}} (-)^{i-1}D(S_i^\alpha S_j^\beta - S_i^\beta S_j^\alpha), \\
&\Omega \sum_j (S_{j-1}^x S_j^y S_{j+1}^x - S_{j-1}^y S_j^x S_{j+1}^y), \\
&\Omega_2 \sum_j (-)^{j-1}(S_{j-1}^x S_j^y S_{j+1}^x + S_{j-1}^y S_j^x S_{j+1}^y),
\end{aligned}
\tag{12}
$$

which will discussed in detail in later sections,

On the other hand, although the five nonsymmorphic cubic groups all lead to SU(2)$_1$ conformal invariance, they still have different low energy properties in the sense that the corresponding nonsymmorphic nonabelian bosonization formulas are different. The expressions of the nonsymmorphic nonabelian bosonization formulas will be explicitly derived for all the five cubic groups.

# 3 Nonsymmorphic cubic $T$ group

In this section, we demonstrate that the nonsymmorphic cubic $T$ group is the smallest nonsymmorphic symmetry group required for the emergent SU(2)$_1$ conformal invariance. We first show that the nonsymmorphic $T$ group indeed leads to an SU(2)$_1$ conformal invariance at low energies. Since the other four nonsymmorphic cubic groups contain the nonsymmorphic $T$ group as a subgroup, it follows that all the five nonsymmorphic cubic groups are able to produce SU(2)$_1$ conformal invariance. Second, we show that if the symmetry group is lowered from cubic to planar, then the SU(2)$_1$ conformal invariance is in general broken. The above two reasonings establish the fact that the nonsymmorphic cubic $T$ group is indeed the minimal one for ensuring SU(2)$_1$ conformal symmetry.

## 3.1 Construction of the nonsymmorphic $T$ group

The cubic $T$ point group is isomorphic to the alternating group $A_4$, which has the following generator-relation representation,

$$
T = <a, b | a^3 = b^2 = (ab)^3 = e>,
\tag{13}
$$

where $a, b$ are the two generators and $e$ is the identity element in the group.

By removing the symmetry operations $\mathcal{T}$ and $R_I I$ from Eq. (4), we consider the following symmetry group $G_T$,

$$G_T = <R_a T_a, R(\hat{x}, \pi), R(\hat{y}, \pi), R(\hat{z}, \pi)> . \tag{14}$$

We are going to show that

$$G_T / <T_{3a}> \cong T . \tag{15}$$

Let's define

$$a' = R_a T_a , \qquad b' = R(\hat{z}, \pi) . \tag{16}$$

Notice that $a'b'(a')^{-1} = R(\hat{x}, \pi)$, and $R(\hat{z}, \pi)R(\hat{x}, \pi) = R(\hat{y}, \pi)$. As a result, $a'$ and $b'$ can be chosen as the generator of $G_T$, i.e.,

$$G_T = <a', b'> . \tag{17}$$

This means that it is enough to prove the following identity

$$G_T / <T_{3a}> = <a', b'> / <T_{3a}> . \tag{18}$$

To prove Eq. (18), we first show that $G_T / <T_{3a}>$ is a subgroup of $T$ by proving that the relations in Eq. (13) are satisfied by $a'$, $b'$ in the sense of modulo $T_{3a}$. To see this, simply notice the following identities,

$$(a')^3 = (a'b')^3 = T_{3a} ,$$
$$(b')^3 = 1 . \tag{19}$$

Then, to prove the isomorphism between $G_T / <T_{3a}>$ and $T$, it is enough to further show that the number of group elements in $G_T / <T_{3a}>$ is no smaller than that of $T$. In fact, there are twelve distinct elements in $G_T$ given by

$$1 = e ,$$
$$R(\hat{x}, \pi) = a'b'(a')^{-1} ,$$
$$R(\hat{y}, \pi) = a'b'(a')^{-1}b' ,$$
$$R(\hat{z}, \pi) = b' ,$$
$$R\left(\frac{1}{\sqrt{3}}(1, 1, 1), -\frac{2\pi}{3}\right) T_a = a' ,$$
$$R\left(\frac{1}{\sqrt{3}}(1, 1, 1), \frac{2\pi}{3}\right) T_{-a} = (a')^{-1} ,$$
$$R\left(\frac{1}{\sqrt{3}}(1, -1, -1), -\frac{2\pi}{3}\right) T_a = a'b'a'b'(a')^{-1} ,$$
$$R\left(\frac{1}{\sqrt{3}}(1, -1, -1), \frac{2\pi}{3}\right) T_{-a} = a'b'(a')^{-1}b'(a')^{-1} ,$$
$$R\left(\frac{1}{\sqrt{3}}(-1, 1, -1), -\frac{2\pi}{3}\right) T_a = a'b'(a')^{-1}b'(a'b')^2(a')^{-1} ,$$
$$R\left(\frac{1}{\sqrt{3}}(-1, 1, -1), \frac{2\pi}{3}\right) T_{-a} = a'[b'(a')^{-1}]^2 b'a'b'(a')^{-1} ,$$
$$R\left(\frac{1}{\sqrt{3}}(-1, -1, 1), -\frac{2\pi}{3}\right) T_a = b'a'(b')^{-1} ,$$
$$R\left(\frac{1}{\sqrt{3}}(-1, -1, 1), \frac{2\pi}{3}\right) T_{-a} = b'(a')^{-1}(b')^{-1} . \tag{20}$$

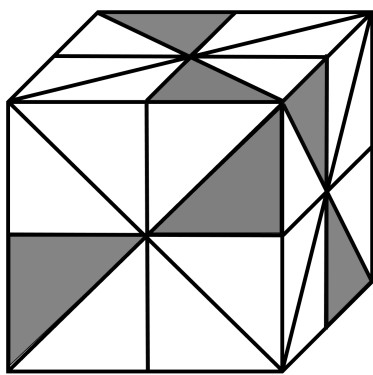

Figure 3: Decorated cube with the symmetry group as the $T$ group.

Since the above twelve operations act in distinct ways in the spin space (as can be seen from the left hand side of the equalities in Eq. (20)), they also act differently in the quotient group $G_T/<T_{3a}>$. Hence we have $|G_T/<T_{3a}>| \geq 12$. On the other hand, $|T| = 12$, and as a result $|G_T/<T_{3a}>| \geq |T|$. Combining with the already proved fact that $G_T/<T_{3a}>$ is a subgroup of $T$, we see that the two must be isomorphic to each other.

We note that there is an intuitive understanding of the isomorphism in Eq. (15). If the spatial components in the symmetry operations in Eq. (20) are temporarily ignored, then it can be observed that all the operations restricted within the spin space leave the decorated cube in Fig. 3 invariant. On the other hand, the symmetry group of the decorated cube in Fig. 3 is the cubic $T$ group, hence it is not a surprise that $G_T$ is intimately related to the cubic $T$ group.

## 3.2 Emergent SU(2)$_1$ conformal symmetry

Having established the isomorphism in Eq. (15), we next prove that $G_T$ is enough to ensure the emergent SU(2)$_1$ conformal invariance. The strategy is to take the 1D spin-1/2 symmetric gamma model in Eq. (3) as the unperturbed system, and then consider all the perturbations allowed by the symmetry group $G_T$. The conclusion of emergent SU(2)$_1$ invariance follows by showing that the low energy field theory up to relevant and marginal terms (in the sense of renormalization group (RG)) remain to be the SU(2)$_1$ WZW model.

In 1+1 dimension, the relevant and marginal terms correspond to the operators having scaling dimensions smaller than and equal to two, respectively. Using the facts that the SU(2)$_1$ WZW current operators and the primary fields have scaling dimensions equal to 1 and 1/2, respectively, the relevant and marginal terms are exhausted by the following terms, which can be analyzed by applying the symmetry transformation properties summarized in Appendix B.

1) The dimension 1/2 operators $\epsilon = \text{tr}(g)$ and $N^\alpha$ ($\alpha = x, y, z$) are forbidden by $T_{3a}$, since $g$ changes sign under $T_{3a}$.

2) The dimension 1 operators $J_\nu^\alpha$ ($\alpha = x, y, z$ and $\nu = L, R$) are forbidden by $R(\hat{\beta}, \pi)$ where $\beta \neq \alpha$, since $J_\nu^\alpha$ changes sign under $R(\hat{\beta}, \pi)$ where $\alpha \neq \beta$ ($\alpha, \beta \in \{x, y, z\}$).

3) The dimension 3/2 operators $J_L^\alpha \epsilon$, $J_R^\alpha \epsilon$, $J_L^\alpha N^\beta$, and $J_R^\alpha N^\beta$ are forbidden by $T_{3a}$, since the signs of $J_\nu^\alpha$ ($N^\alpha$) remains unchanged (changed) under $T_{3a}$.

4) The dimension 2 operators are in general of the forms $J_L^\alpha J_L^\beta$, $J_R^\alpha J_R^\beta$, and $J_L^\alpha J_R^\beta$, where $\alpha, \beta \in \{x, y, z\}$. In the continuum limit, the translation operator $T_a$ becomes an internal symmetry, which acts as identity on $J_L^\alpha$ and $J_R^\alpha$. There are four irreducible representations of the $T$ group, given by $A$, $E_1$, $E_2$ and $T$ [42] (note: whether the symbol $T$ refers to the representation $T$ or the cubic group $T$ should be clear from the context). Both span$\{J_L^\alpha | \alpha = x, y, z\}$ and span$\{J_R^\alpha | \alpha = x, y, z\}$ correspond to the $T$ representation, which is three-dimensional, where

span{...} represents the vector space spanned by the elements within the bracket. Using the tensor product rule $T \otimes T = A \oplus E_1 \oplus E_2 \oplus T$, we see that the only terms which are invariant under the $T$ group (i.e., corresponding to the $A$ representation) are $\vec{J}_L \cdot \vec{J}_L$, $\vec{J}_R \cdot \vec{J}_R$ and $\vec{J}_L \cdot \vec{J}_R$.

Based on the above analysis, the low energy Hamiltonian compatible with the nonsymmorphic cubic $T$ group is

$$\mathcal{H}_1 = \mathcal{H}_1^{(0)} - u \int dx \vec{J}_L \cdot \vec{J}_R, \tag{21}$$

in which

$$\begin{aligned}
\mathcal{H}_1^{(0)} &= \int dx \frac{2\pi}{3} (v_L \vec{J}_L \cdot \vec{J}_L + v_R \vec{J}_R \cdot \vec{J}_R) \\
&= \int dx \frac{2\pi}{3} v (\lambda \vec{J}_L \cdot \vec{J}_L + \lambda^{-1} \vec{J}_R \cdot \vec{J}_R),
\end{aligned} \tag{22}$$

where $v = \sqrt{v_L v_R}$, and $\lambda = \sqrt{v_L/v_R}$. Notice that because of a lack of time reversal and inversion symmetries, the velocities of the left and right movers are in general different, which is different from the case of nonsymmorphic $O_h$ symmetry in Eq. (7). We will absorb the velocity $v$ into a redefinition of time in what follows, or effectively, $v = 1$.

Next we show that the Hamiltonian $\mathcal{H}_1$ in Eq. (21) has an emergent SU(2)$_1$ conformal symmetry at low energies when $u > 0$, i.e., a positive $u$ is an irrelevant operator in the RG sense. The strategy is to study the one-loop RG flow of the coupling $u$, by taking $\mathcal{H}_1^{(0)}$ as the unperturbed system. To facilitate the RG analysis, we will take the following more general version of the Hamiltonian [40]

$$\mathcal{H}_1' = \mathcal{H}_1^{(0)} - \sum_{\alpha=x,y,z} u_\alpha \int dx J_L^\alpha J_R^\alpha, \tag{23}$$

such that Eq. (21) corresponds to the special case $u_\alpha \equiv u$ in Eq. (23).

We use the standard method of operator product expansion (OPE) to derive the one-loop RG flow equations [39,40] for $v$, $\lambda$, and $u$. In imaginary time, all the information of the system is encoded in the following time ordered product,

$$T \left( e^{-\int d\tau dx \sum_{\alpha=x,y,z} (-u_\alpha J_L^\alpha J_R^\alpha)} \right), \tag{24}$$

in which $J_L^\alpha$ and $J_R^\alpha$ are the operators in the interaction picture defined by the unperturbed Hamiltonian $\mathcal{H}_1^{(0)}$, i.e.,

$$\begin{aligned}
J_L^\alpha(\tau, x) &= e^{\tau \mathcal{H}_1^{(0)}} J_L(\tau=0, x) e^{-\tau \mathcal{H}_1^{(0)}} = J_L^\alpha(z), \\
J_R^\alpha(\tau, x) &= e^{\tau \mathcal{H}_1^{(0)}} J_R(\tau=0, x) e^{-\tau \mathcal{H}_1^{(0)}} = J_R^\alpha(\bar{z}'),
\end{aligned} \tag{25}$$

where

$$\begin{aligned}
z &= \lambda \tau + ix, \\
\bar{z}' &= \lambda^{-1} \tau - ix.
\end{aligned} \tag{26}$$

Notice that since $\lambda$ can be different from 1, $\bar{z}'$ may not be the complex conjugate of $z$. According to the chiral SU(2)$_1$ WZW theory, the operator product expansions (OPE) of $J_L^\alpha$ and $J_R^\alpha$ are given by [43]

$$J_L^\alpha(z) J_L^\beta(w) = \frac{\delta_{\alpha\beta}}{8\pi^2(z-w)^2} + \frac{i\epsilon_{\alpha\beta\gamma} J_L^\gamma(w)}{2\pi(z-w)} + (J_L^\alpha J_L^\beta)(w) + O(z-w),$$

$$J_R^\alpha(\bar{z}') J_R^\beta(\bar{w}') = -\frac{\delta_{\alpha\beta}}{8\pi^2(\bar{z}'-\bar{w}')^2} + \frac{i\epsilon_{\alpha\beta\gamma} J_L^\gamma(\bar{w}')}{2\pi(\bar{z}'-\bar{w}')} + (J_R^\alpha J_R^\beta)(\bar{w}') + O(\bar{z}'-\bar{w}'), \tag{27}$$

in which $(AB)(w)$ represents the normal ordered product of the OPE $A(z)B(w)$, i.e., the $O(1)$ term in the Laurent expansion of $A(z)B(w)$ in terms of $z - w$, and similarly for $(AB)(\bar{w}')$ in the anti-holomorphic sector.

Expanding Eq. (24) to second order, we obtain

$$1 + \int d\tau dx \sum_{\alpha=x,y,z} u_\alpha J_L^\alpha J_R^\alpha$$
$$+ \frac{1}{2} \int_a d\tau_1 dx_1 d\tau_2 dx_2 \sum_{\alpha=x,y,z} \sum_{\beta=x,y,z} u_\alpha u_\beta J_L^\alpha(z_1) J_R^\alpha(\bar{z}_1') J_L^\beta(z_2) J_R^\beta(\bar{z}_2') + \dots, \quad (28)$$

in which $z$ and $\bar{z}'$ are given in Eq. (26); $\tau_-$, $x_-$, $z_-$, $\bar{z}_-$ are

$$\tau_- = \tau_1 - \tau_2,$$
$$x_- = x_1 - x_2,$$
$$z_- = \lambda\tau_- + ix_-,$$
$$\bar{z}_-' = \lambda^{-1}\tau_- - ix_-, \quad (29)$$

and $\int_a$ indicates that the integration is subject to a real space cutoff $a$, i.e., the integration range is restricted to

$$\sqrt{|\tau_1 - \tau_2|^2 + |x_1 - x_2|^2} \geq a. \quad (30)$$

To perform RG, we increase the real space cutoff from $a$ to $ba$, by integrating over the fields within the range

$$a \leq \sqrt{|\tau_1 - \tau_2|^2 + |x_1 - x_2|^2} \leq ba. \quad (31)$$

Using the OPE in Eq. (27) and integrating over the modes in Eq. (31), the second order term in Eq. (28) contains the following terms

$$\frac{1}{2} \int_{ba} d\tau dx \int_a^{ba} d\tau_- dx_- \frac{1}{8\pi^2(z_-)^2} \sum_{\alpha=x,y,z} (u_\alpha)^2 (J_R^\alpha J_R^\alpha)(\bar{z}')$$
$$- \frac{1}{2} \int_{ba} d\tau dx \int_a^{ba} d\tau_- dx_- \frac{1}{8\pi^2(\bar{z}_-')^2} \sum_{\alpha=x,y,z} (u_\alpha)^2 (J_L^\alpha J_L^\alpha)(z)$$
$$- \frac{1}{2} \int_{ba} d\tau dx \int_a^{ba} d\tau_- dx_- \frac{1}{4\pi^2 z_- \bar{z}_-'} \sum_{\alpha=x,y,z} \sum_{\beta=x,y,z} \sum_{\gamma=x,y,z} (\epsilon_{\alpha\beta\gamma})^2 u_\alpha u_\beta J_L^\gamma(z) J_R^\gamma(\bar{z}'), \quad (32)$$

in which $\int_a^{ba}$ means that the integration is restricted within the range in Eq. (31). The integrations $\int_a^{ba} d\tau_- dx_-$ can be evaluated as

$$\int_a^{ba} d\tau_- dx_- \frac{1}{(z_-)^2} = 0,$$
$$\int_a^{ba} d\tau_- dx_- \frac{1}{(\bar{z}_-')^2} = 0,$$
$$\int_a^{ba} d\tau_- dx_- \frac{1}{z_- \bar{z}_-'} = \ln b \frac{4\pi}{\sqrt{4 + (\lambda^{-1} - \lambda)^2}}. \quad (33)$$

Hence Eq. (32) reduces to

$$\sum_{\gamma=x,y,z} [-\ln b \frac{1}{4\pi\sqrt{1 + (\lambda^{-1} - \lambda)^2/4}} \sum_{\alpha,\beta=x,y,z} (\epsilon_{\alpha\beta\gamma})^2 u_\alpha u_\beta] \int dx J_L^\gamma J_R^\gamma. \quad (34)$$

Clearly, there is no renormalization of $v$ and $\lambda$, i.e.,

$$\frac{dv}{d\ln b} = 0,$$
$$\frac{d\lambda}{d\ln b} = 0, \tag{35}$$

but there is a renormalization of $u_\alpha$.

Since the tree level scaling for $u_\alpha$ vanishes (as $J_L^\alpha J_R^\alpha$ is marginal where $\alpha = x, y, z$), we obtain the following one-loop RG flow equations,

$$\frac{du_x}{d\ln b} = -\frac{u_y u_z}{2\pi\sqrt{1 + (\lambda^{-1} - \lambda)^2/4}},$$
$$\frac{du_y}{d\ln b} = -\frac{u_z u_x}{2\pi\sqrt{1 + (\lambda^{-1} - \lambda)^2/4}},$$
$$\frac{du_z}{d\ln b} = -\frac{u_x u_y}{2\pi\sqrt{1 + (\lambda^{-1} - \lambda)^2/4}}, \tag{36}$$

in which a factor of two is included since both $(\epsilon_{\alpha\beta\gamma})^2 u_\alpha u_\beta$ and $(\epsilon_{\beta\alpha\gamma})^2 u_\beta u_\alpha$ contribute to the renormalization of $u_\gamma$. In the special case $u_\alpha \equiv u$, Eq. (36) becomes

$$\frac{du}{d\ln b} = -\frac{u^2}{2\pi\sqrt{1 + (\lambda^{-1} - \lambda)^2/4}}. \tag{37}$$

It is clear from Eq. (37) that $u$ is marginally irrelevant (relevant) when $u > 0$ ($u < 0$). When $\lambda = 1$, Eq. (37) reduces to the standard RG flow equations for $v_L = v_R$ in Ref. [40].

Hence, in the extreme infrared limit when $u > 0$, the low energy Hamiltonian flows to $\mathcal{H}_1^{(0)}$ in Eq. (22). That is to say, the system has an emergent $SU(2)_1$ conformal invariance, but with different velocities for the left and right movers. Notice that since $u$ is indeed positive for the symmetric gamma model (as this model is numerically verified to have emergent $SU(2)_1$ conformal symmetry), it must remain positive at least when the perturbations are small enough. That is to say, for the models with a nonsymmorphic cubic $T$ symmetry group, there exists an extended region in the phase diagram which has an emergent $SU(2)_1$ conformal invariance at low energies.

By closing this subsection, we derive the nonsymmorphic nonabelian bosonization formulas which are consistent with the nonsymmorphic $T$ symmetry group. Since $\{1, R(\hat{x}, \pi), R(\hat{y}, \pi), R(\hat{z}, \pi)\}$ ($\cong \mathbb{Z}_2 \times \mathbb{Z}_2$) is a subgroup of the symmetry group $G_T$, there is no cross-directional terms in the bosonization formulas, i.e., $S_j^\alpha$ does not contain $J^\beta$, $N^\beta$ where $\beta \neq \alpha$. Requiring the left and right hand sides of Eq. (9) to be covariant under the nonsymmorphic $T$ group, we obtain

$$D_{v,1}^z = D_{v,2}^y = D_{v,3}^x = D_1^{(v)},$$
$$D_{v,1}^x = D_{v,2}^z = D_{v,3}^y = D_2^{(v)},$$
$$D_{v,1}^y = D_{v,2}^x = D_{v,3}^z = D_3^{(v)}, \tag{38}$$

and

$$C_1^z = C_2^y = C_3^x = C_1,$$
$$C_1^x = C_2^z = C_3^y = C_2,$$
$$C_1^y = C_2^x = C_3^z = C_3, \tag{39}$$

in which $\nu = L, R$. We note that since there is no time reversal nor inversion symmetry, in general $D_\mu^{(L)} \neq D_\mu^{(R)}$ ($\mu = 1, 2, 3$). Eqs. (38,39) reduce back to the nonsymmorphic $O_h$ case in Eqs. (10,11) by imposing the conditions $D_\mu^{(L)} = D_\mu^{(R)} = D_\mu$, $D_2 = D_3$, and $C_2 = C_3$.

The $SU(2)_1$ WZW model combined with the nonsymmorphic nonabelian bosonization formulas enable the derivations of the spin correlation functions. Under proper normalizations, the $SU(2)_1$ WZW model predicts

$$\langle J_L^\alpha(z) J_L^\beta(w) \rangle = \delta_{\alpha\beta} \frac{1}{(z-w)^2} ,$$
$$\langle J_R^\alpha(\bar{z}') J_R^\beta(\bar{w}') \rangle = \delta_{\alpha\beta} \frac{1}{(\bar{z}'-\bar{w}')^2} ,$$
$$\langle N^\alpha(z,\bar{z}') N^\beta(w,\bar{w}') \rangle = \delta_{\alpha\beta} \frac{1}{\sqrt{(z-w)(\bar{z}'-\bar{w}')}} . \tag{40}$$

In the static case, the above equations can be simplified into

$$\langle J_L^\alpha(r+x) J_L^\beta(r) \rangle = -\frac{\delta_{\alpha\beta}}{x^2} ,$$
$$\langle J_R^\alpha(r+x) J_R^\beta(r) \rangle = -\frac{\delta_{\alpha\beta}}{x^2} ,$$
$$\langle N^\alpha(r+x) N^\beta(r) \rangle = \frac{\delta_{\alpha\beta}}{|x|} , \tag{41}$$

in which both $x$ and $r$ are spatial coordinates and the time $\tau$ is implicitly set to zero. Then combining with the nonsymmorphic nonabelian bosonization formulas, we obtain the following static spin correlation functions $S^{\alpha\alpha}(r) = \langle S_1^\alpha S_{1+r}^\alpha \rangle$ as

$$S^{\alpha\alpha}(r) = S_0^{\alpha\alpha}(r) + (-)^r S_\pi^{\alpha\alpha}(r) + \sin\left(\frac{\pi}{3}r\right) S_{\pi/3,(1)}^{\alpha\alpha}(r) + \cos\left(\frac{\pi}{3}r\right) S_{\pi/3,(2)}^{\alpha\alpha}(r)$$
$$+ \sin\left(\frac{2\pi}{3}r\right) S_{2\pi/3,(1)}^{\alpha\alpha}(r) + \cos\left(\frac{2\pi}{3}r\right) S_{\pi/3,(2)}^{\alpha\alpha}(r), \tag{42}$$

in which $S_\pi^{\alpha\alpha}(r)$ ($\alpha = x, y, z$) is given by

$$S_\pi^{\alpha\alpha}(r) = A_\alpha \frac{\ln^{1/2}(r/r_0)}{r} , \tag{43}$$

where

$$A_x = \frac{1}{3}[(C_2)^2 + C_2 C_3 + C_2 C_1],$$
$$A_y = \frac{1}{3}[(C_3)^2 + C_3 C_1 + C_3 C_2],$$
$$A_z = \frac{1}{3}[(C_1)^2 + C_1 C_2 + C_1 C_3]. \tag{44}$$

We note that the logarithmic factor in Eq. (43) arises from the marginally irrelevant operator $\vec{J}_L \cdot J_R$ in the low energy field theory in Eq. (21). For finite size periodic systems, $r$ should be replaced by $r_L = \frac{L}{\pi} \sin(\frac{\pi r}{L})$ according to conformal field theory on cylinders. The expressions of other five Fourier components in Eq. (42) are included in Appendix C.

### 3.3 $T$ as the smallest group realizing $SU(2)_1$ conformal invariance

Finally, we argue that if the symmetry group is lowered from cubic to planar, then in general the emergent $SU(2)_1$ conformal invariance will be lost. Let's consider the dimension 2

operators $J_L^\alpha J_R^\beta$. Without loss of generality, let's also assume that the system has time reversal symmetry, since if the $SU(2)_1$ conformal invariance is already lost in the presence of time reversal symmetry, it must also be lost when time reversal symmetry is absent. Since time reversal switches the left and right movers, it requires the symmetric combination $J_L^\alpha J_R^\beta + J_L^\beta J_R^\alpha$. For such symmetric combinations, the spatial inversion can also be effectively viewed as an internal symmetry acting as identity, since in view of representations of inversion symmetry, $J_L^\alpha J_R^\beta + J_L^\beta J_R^\alpha$ has no difference from $J^\alpha J^\beta$, where spatial inversion acts trivially on the vector space $\mathrm{span}\{J^\alpha | \alpha = x, y, z\}$.

Hence, we see that for the quadratic terms $J_L^\alpha J_R^\beta + J_L^\beta J_R^\alpha$, the action of the nonsymmorphic symmetry group effectively reduces to a subgroup of $SU(2)$, and again in view of representations, there is no difference to consider $J^\alpha J^\beta$, where $\mathrm{span}\{J^\alpha | \alpha = x, y, z\}$ is a vector representation (i.e., angular momentum 1) of the $SU(2)$ group. Using the angular momentum addition rule, we have $1 \otimes 1 = 0 \oplus 1 \oplus 2$, where "$n$" ($n \in \mathbb{Z}$) represents the representation of the $SU(2)$ group with the value of the angular momentum equal to $n$. Keeping only the symmetric combinations, this becomes $1 \otimes 1 = 0 \oplus 2$. When the symmetry group is lowered from cubic to planar, we should consider $U(1)$ instead of $SU(2)$, i.e., the planar nonsymmorphic group effectively acts as a subgroup of $U(1)$. In the planar case, the quintet sector (i.e., the sector of angular momentum 2) can be further decomposed, which contains an $L^z = 0$ state, where $L^z$ is the quantum number for the $U(1)$ group. Notice that this state is not invariant under cubic symmetry groups, since the decomposition of the quintet sector according to cubic groups is in general $2 = E \oplus T$ which does not contain any one-dimensional irreducible representation, where $E$ and $T$ represent the two- and three-dimensional irreducible representations of the cubic groups.

In this way, we see that the low energy field theory for a planar nonsymmorphic group is in general at most of the XXZ type (i.e., at most having emergent $U(1)$ symmetry), thereby spoiling the emergent $SU(2)_1$ conformal symmetry. We note that there may be other operators with scaling dimensions smaller than 2 which are allowed by more general nonsymmorphic planar groups. However, since $SU(2)_1$ conformal invariance is already broken at the level of dimension 2 operators, it must also be broken in more general cases where additional operators are allowed by symmetries.

Based on the above analysis, we conclude that the nonsymmorphic cubic $T$ group is the smallest group required for an emergent $SU(2)_1$ conformal symmetry at low energies. Here we emphasize that it is still possible for the system to have emergent $SU(2)_1$ conformal invariance for planar nonsymmorphic groups under special circumstances, for example, at the continuous phase transition points between Luttinger liquid and ordered phases (i.e., a transition from planar XXZ to axial XXZ models) [32]. However, in this case, the region having emergent $SU(2)_1$ conformal invariance does not occupy an extended volume in the phase diagram, or to say, such region has zero measure and requires fine tuning. On the other hand, the $SU(2)_1$ conformal invariance ensured by nonsymmorphic cubic group symmetries is a generic symmetry property of the model, not requiring any fine tuning.

## 3.4 The asymmetric-gamma-octupole model

We consider the following "asymmetric-gamma-octupole model"

$$H_{A\Gamma\Omega} = H_{A\Gamma} + \Omega\left(\sum_j S_{j-1}^x S_j^y S_{j+1}^x - \sum_j S_{j-1}^y S_j^x S_{j+1}^y\right), \tag{45}$$

in which the asymmetric gamma term $H_{A\Gamma}$ is defined as

$$H_{A\Gamma} = H_{S\Gamma} + \sum_{<ij>\in\gamma \text{ bond}} (-)^{i-1} D\left(S_i^\alpha S_j^\beta - S_i^\beta S_j^\alpha\right), \tag{46}$$

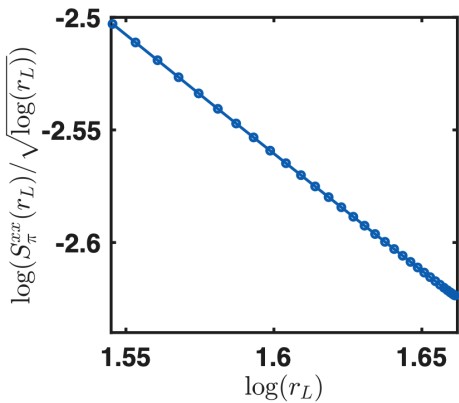

Figure 4: $S_\pi^{xx}(r_L)/\ln^{1/2}(r_L)$ as a function of $r_L$ on a log-log scale where the slope is $-1.042$, in which $r_L = \frac{L}{\pi}\sin(\frac{\pi r}{L})$. DMRG numerics are performed for the asymmetric-gamma-octupole model in the $U_6$ frame on a system of $L = 144$ sites using periodic boundary conditions. The parameters are chosen as $\Gamma_1 = -0.8$, $\Gamma_2 = -1.2$, $\Omega = 0.3$.

and the $\Omega$ term is a spin-octupolar term. We note that the $D$ term in Eq. (46) is a staggered site-dependent Dzyaloshinskii-Moriya interaction which can be generated by a staggered electric field along $z$-direction.

Similarly, $H'_{A\Gamma\Omega}$ in the six-sublattice rotated frame is defined as $H'_{A\Gamma\Omega} = (U_6)^{-1}H_{A\Gamma\Omega}U_6$, given by

$$H'_{A\Gamma\Omega} = \sum_{<ij>\in\gamma\text{ bond}} \left(-\Gamma_1 S_i^\alpha S_j^\alpha - \Gamma_2 S_i^\beta S_j^\beta\right) + \Omega\sum_j \left(S_{j-1}'^{\alpha_l} S_j'^{\gamma_l} S_{j+1}'^{\beta_l} - S_{j-1}'^{\alpha_r} S_j'^{\gamma_r} S_{j+1}'^{\beta_r}\right), \qquad (47)$$

in which $\Gamma_1 = \Gamma + D$, $\Gamma_2 = \Gamma - D$; $\gamma = x, z, y$ has a three-site periodicity as shown in Fig. 1 (b); $\gamma_l = <j-1, j>$ and $(\alpha_l, \beta_l, \gamma_l)$ form a right-handed coordinate system; $\gamma_r = <j, j+1>$ and $(\alpha_r, \beta_r, \gamma_r)$ form a right-handed coordinate system. The explicit expressions of $H_{A\Gamma\Omega}$ and $H'_{A\Gamma\Omega}$ are included in Appendix A. It can be verified that $H'_{A\Gamma\Omega}$ is invariant under all the symmetry operations in Eq. (4) except $\mathcal{T}$ and $R_I I$. Hence the symmetry group $G_{A\Gamma\Omega}$ satisfies

$$G_{A\Gamma\Omega} = G_T, \qquad (48)$$

where $G_T$ is defined in Eq. (14). This shows that $H_{A\Gamma\Omega}$ provides a concrete realization for the nonsymmorphic $T$ group, and we expect that the system has an emergent SU(2)$_1$ conformal symmetry at low energies for a range of nonzero $D$ and $\Omega$. We note that there are many other terms which preserve the nonsymmorphic $T$ symmetry, and Eq. (45) is only one of the many possibilities.

We discuss the numerical evidence for the emergent SU(2)$_1$ invariance by comparing numerical results with the predictions in Eq. (43). Fig. 4 shows the numerical results of $S_\pi^{xx}(r_L)/\ln^{1/2}(r_L)$ as a function of $r_L$ on a log-log scale for the asymmetric-gamma-octupole model $H'_{A\Gamma\Omega}$ in Eq. (47) in the $U_6$ frame at $\Gamma_1 = -0.8$, $\Gamma_2 = -1.2$, and $\Omega = 0.3$, obtained from DMRG simulations on a system of $L = 144$ sites using periodic boundary conditions, in which $r_L = \frac{L}{\pi}\sin(\frac{\pi r}{L})$ in accordance with conformal field theory on finite size systems. The slope extracted from Fig. 4 (a) is $-1.042$, which is very close to $-1$, consistent with the prediction of the SU(2)$_1$ WZW model in Eq. (43).

## 3.5 Numerical evidence for velocity difference in left and right chiral sectors

In Eq. (22), we have proved that because of a lack of inversion and time reversal symmetries, for the case of the nonsymmorphic $T$ group, the velocities in the left and right chiral sectors have different values. Here we numerically demonstrate such velocity difference.

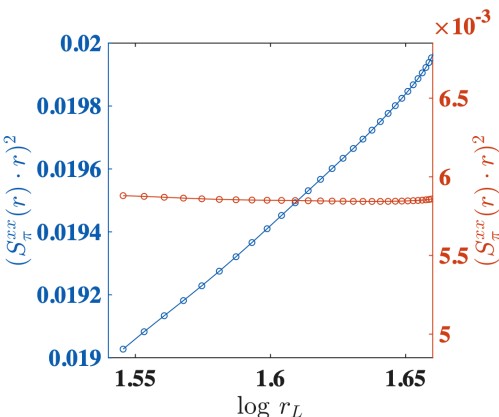

Figure 5: $[r_L S_\pi^{xx}(r_L)]^2$ versus $\log(r_L)$ for $J_2 = 0$ (blue points) and $J_2 = 0.13$ (orange points), in which $r_L = \frac{L}{\pi}\sin(\frac{\pi r}{L})$. DMRG numerics are performed for $H_{A\Gamma\Omega}^{(2)}$ defined in Eq. (55) on a system of $L = 144$ sites using periodic boundary conditions.

In general, the left and right sectors are coupled by the marginal term $u \int dx \vec{J}_L \cdot \vec{J}_R$ in Eq. (21). To decouple the two chiral sectors, we use a trick by adding a next-nearest neighboring Heisenberg term. The model that we consider is the following "asymmetric-gamma-octupole-$J_2$ model",

$$H_{A\Gamma\Omega}^{(2)} = H_{A\Gamma\Omega} + J_2 \sum_i \vec{S}_i \cdot \vec{S}_{i+2}\,. \tag{49}$$

In the low energy field theory, the $J_2$ term renormalizes the marginal coupling $u$ in Eq. (21). At a critical value $J_{2c}$, the coupling $u$ vanishes, and the logarithmic correction in the correlation functions at $J_2 = J_{2c}$ disappear. In Fig. 5, $[r_L S_\pi^{xx}(r_L)]^2$ as a function of $\log(r_L)$ is plotted by the blue hollow circles, in which the parameters are taken as $\Gamma_1 = -0.8$, $\Gamma_2 = -1.2$, $\Omega = 0.3$, and $J_2 = 0$, the same as those in Fig. 4. Clearly, the approximately linear relation between $[r_L S_\pi^{xx}(r_L)]^2$ and $\log(r_L)$ is consistent with the prediction of the logarithmic correction in Eq. (43). When $J_2 = 0.13$ is further added to the Hamiltonian, the numerical results of $[r_L S_\pi^{xx}(r_L)]^2$ as a function of $\log(r_L)$ become the orange hollow circles in Fig. 5, which is approximately flat with a vanishing slope, indicating an absence of the logarithmic correction and a critical value $J_{2c}$ close to 0.13.

Next, we consider periodic chains with odd lengths, where $J_2$ is taken as $J_{2c}$ such that the two chiral sectors are decoupled. As discussed in Ref. [44] (see Table I therein), when the velocities in the left and right chiral sectors are the same, the energy spectrum for systems with an SU(2)$_L$×SU(2)$_R$ symmetry exhibits a four-fold ground state degeneracy for odd periodic chains, two coming from the left chiral sector and the other two coming from the right chiral sector, which is numerically confirmed for the spin-1/2 $J_1$-$J_2$ Heisenberg chain with system size $L = 19$ shown in Fig. 6 in Ref. [44].

On the other hand, when the velocities are different, it is expected that the four-fold ground state degeneracy is split into two groups, each having a two-fold degeneracy. Fig. 6 (a) shows the energies $E - E_0$ of the four lowest states at different odd system sizes $L$ with periodic boundary conditions, where the energies are measured from the ground state energy $E_0$ of the corresponding system size. The parameters are taken as $\Gamma_1 = -0.8$, $\Gamma_2 = -1.2$, $\Omega = 0.3$, and $J_2 = 0.13$ in DMRG numerical calculations. The splitting of the four states into two degenerate groups can be clearly seen in the figure. Furthermore, according to Ref. [44], the energy splitting between the lowest energy states in the two chiral sectors is predicted to be $\frac{\pi(v_L - v_R)}{4L}$, where $v_L$ and $v_R$ are the velocities in the left and right chiral sectors, respectively. In Fig. 6 (b), $E_2(L) - E_0(L)$ versus $L$ are plotted on a log-log scale, where $E_2$ and $E_0$ are the

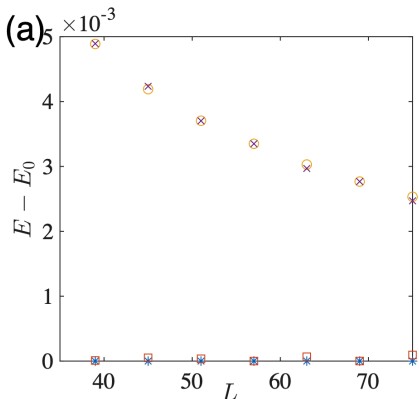
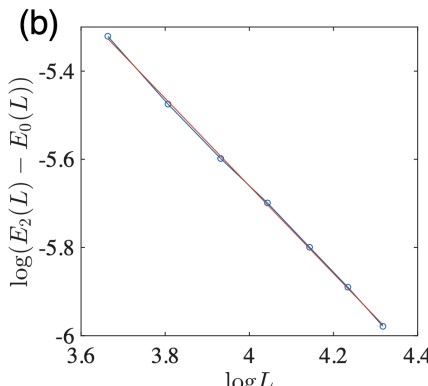

Figure 6: Energies $E - E_0$ of the four lowest states (denoted by the symbols "star", "square", "circle", and "cross") measured from the ground state energy $E_0$ as a function of the system size $L$ (a) on linear scale, and (b) on a log-log scale. DMRG numerics are performed for $H_{A\Gamma\Omega}^{(2)}$ defined in Eq. (49) at $\Gamma_1 = -0.8$, $\Gamma_2 = -1.2$, $\Omega = 0.3$, and $J_2 = 0.13$, with periodic boundary conditions.

energies of the third and first lowest states, respectively. As can be seen from Fig. 6 (b), the relation is linear with a slope $-1$, consistent with the prediction $\lambda L^{-1}$ where $\lambda = \pi(\nu_L - \nu_R)/4$.

# 4 Nonsymmorphic cubic $T_h$ group

In this section, the nonsymmorphic $T_h$ group is discussed. We construct the nonsymmorphic $T_h$ group, give the minimal model with nonsymmorphic $T_h$ symmetry, and present numerical evidence obtained from DMRG simulations.

## 4.1 Construction of the nonsymmorphic $T_h$ group

The cubic $T_h$ group contains 24 group elements. In the language of crystalline point groups, the $T_h$ group can be obtained from the $T$ group by including the spatial inversion operation, i.e., $T_h \cong T \times \mathbb{Z}_2$. In our case, time reversal operation $\mathcal{T}$ plays the role of inversion since $\mathcal{T}$ changes the sign of the spin operators. The set of generators of $G_{T_h}$ can be obtained from Eq. (17) by adding $\mathcal{T}$, as

$$G_{T_h} = <\mathcal{T}, R_a T_a, R(\hat{z}, \pi)>, \tag{50}$$

which satisfies

$$G_{T_h}/<T_{3a}> \cong T_h. \tag{51}$$

We note that there is an intuitive understanding of the isomorphism in Eq. (51). If the spatial components in the symmetry operations in Eq. (50) are temporarily ignored, then it can be observed that all the operations restricted within the spin space leave the decorated cube in Fig. 7 invariant. On the other hand, the symmetry group of the decorated cube in Fig. 7 is the cubic $T_h$ group, hence it is not a surprise that $G_{T_h}$ is intimately related to the cubic $T_h$ group. Notice that Fig. 7 has a larger symmetry group than Fig. 3, since the former is also invariant under inversion (corresponding to time reversal) while the latter is not.

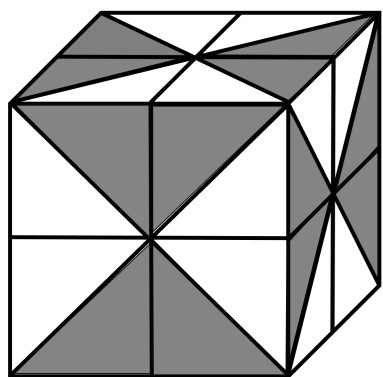

Figure 7: Decorated cube with the symmetry group as the $T_h$ group.

Next, we derive the nonsymmorphic nonabelian bosonization formulas which are consistent with the nonsymmorphic $T_h$ group. Since $T$ is a subgroup of $T_h$, the relations in Eqs. (38,39) also apply to the $T_h$ case. Further imposing the time reversal symmetry, we obtain the following constraints for the bosonization coefficients

$$
\begin{aligned}
D_1^{(L)} &= D_1^{(R)} = D_1\,, \\
D_2^{(L)} &= D_2^{(R)} = D_2\,, \\
D_3^{(L)} &= D_3^{(R)} = D_3\,,
\end{aligned}
\tag{52}
$$

in addition to those in Eqs. (38,39). Using the nonsymmorphic bosonization formulas and the $SU(2)_1$ conformal field theory, the component $S_\pi^{\alpha\alpha}(r)$ of the spin correlation function $\langle S_1^\alpha S_{1+r}^\alpha \rangle$ ($\alpha = x, y, z$) in the $U_6$ frame can be shown to be same as the case of the cubic $T$ group given in Eq. (43).

## 4.2 The asymmetric gamma model

We consider the "asymmetric Gamma model" $H_{A\Gamma}$ defined in Eq. (46). More explicitly, the Hamiltonian can be written in the following form,

$$
H_{A\Gamma} = \sum_n (\Gamma_1 S_{2n-1}^y S_{2n}^z + \Gamma_2 S_{2n-1}^z S_{2n}^y) + \sum_n (\Gamma_1 S_{2n}^x S_{2n+1}^z + \Gamma_2 S_{2n-1}^z S_{2n+1}^x)\,,
\tag{53}
$$

in which $\Gamma = (\Gamma_1 + \Gamma_2)/2$, $D = (\Gamma_1 - \Gamma_2)/2$. In what follows, we sometimes parametrize $\Gamma_1$ and $\Gamma_2$ as

$$
\Gamma_1 = \cos(\theta)\,, \qquad \Gamma_2 = \sin(\theta)\,.
\tag{54}
$$

Clearly, when $\Gamma_1 = \Gamma_2$, Eq. (53) reduces to the symmetric gamma model defined in Eq. (1). Performing the six-sublattice rotation $U_6$ defined in Eq. (2), $H_{A\Gamma}$ becomes $H'_{A\Gamma} = (U_6)^{-1} H_{A\Gamma} U_6$, given by

$$
H'_{A\Gamma} = \sum_{<ij>\in\gamma\,\text{bond}} \left( -\Gamma_1 S_i^\alpha S_j^\alpha - \Gamma_2 S_i^\beta S_j^\beta \right)\,,
\tag{55}
$$

in which $\gamma = x, z, y$ has a three-site periodicity as shown in Fig. 1 (b). Explicit expressions of $H_{A\Gamma}$ and $H'_{A\Gamma} = (U_6)^{-1} H_{A\Gamma} U_6$ are included in Appendix A.

It can be verified that when $\Gamma_1 \neq \Gamma_2$, all the symmetries in Eq. (4) remain to be the symmetries of $H'_{A\Gamma}$ except $R_I I$. Hence the symmetry group $G_{A\Gamma}$ of $H'_{A\Gamma}$ is

$$
G_{A\Gamma} = <\mathcal{T}, R_a T_a, R(\hat{x}, \pi), R(\hat{y}, \pi), R(\hat{z}, \pi)>\,.
\tag{56}
$$

Comparing with $G_T$ in Eq. (14), we see that $G_{A\Gamma}$ has an additional time reversal symmetry. Thus, as discussed in Sec. 4.1, we have

$$G_{A\Gamma} = G_{T_h}, \tag{57}$$

i.e., the asymmetric gamma model has a nonsymmorphic $T_h$ symmetry, and it is expected that there is an extended region in the phase diagram of the asymmetric gamma model which has an emergent $SU(2)_1$ conformal symmetry.

Before proceeding on presenting numerical evidences for the asymmetric gamma model, we make some comments on the properties of this model. We first discuss the unitarily equivalent relations in the asymmetric model. For convenience, we work with the unrotated frame and consider $H_{A\Gamma}$.

First, notice that a global spin rotation $R(\hat{z}, \pi)$ around $z$-axis by $\pi$ changes the signs of both $\Gamma_1$ and $\Gamma_2$, hence there is the equivalent relation

$$(\Gamma_1, \Gamma_2) \simeq (-\Gamma_1, -\Gamma_2), \tag{58}$$

i.e., $\theta \simeq \pi + \theta$ up to a unitary transformation. Second, spatial inversion with respect to the middle point of a bond switches $\Gamma_1$ and $\Gamma_2$, hence

$$(\Gamma_1, \Gamma_2) \simeq (\Gamma_2, \Gamma_1), \tag{59}$$

i.e., $\theta \simeq \pi/2 - \theta$. Third, it can checked that by performing $R(\hat{y}, \pi)$ on odd sites and $R(\hat{x}, \pi)$ on even sites, $\Gamma_1$ is sent to $-\Gamma_1$ whereas $\Gamma_2$ remains unchanged, hence there is the equivalence

$$(\Gamma_1, \Gamma_2) \simeq (-\Gamma_1, \Gamma_2), \tag{60}$$

i.e., $\theta \simeq \pi - \theta$. Based on the above discussions, we see that it is enough to consider the parameter region $\theta \in [\pi/4, \pi/2]$.

Another interesting property is that the asymmetric gamma model is exactly solvable via a Jordan-Wigner transformation when one of $\Gamma_1$ and $\Gamma_2$ vanishes. Because of the equivalent relations in Eqs. (58,59,60), it is enough to consider the case $\Gamma_2 = 0$, $\Gamma_1 > 0$. Then the model becomes

$$H_{\Gamma_1} = \Gamma_1 \sum_n (S^y_{2n-1} S^z_{2n} + S^x_{2n} S^z_{2n+1}). \tag{61}$$

In fact, by the following two-sublattice unitary transformations $V_2$,

$$\begin{aligned}
&\text{Sublattice 1: } (S^x_{2n-1}, S^y_{2n-1}, S^z_{2n-1}) \rightarrow (-S^z_{2n-1}, -S^y_{2n-1}, -S^x_{2n-1}), \\
&\text{Sublattice 2: } (S^x_{2n}, S^y_{2n}, S^z_{2n}) \quad\quad \rightarrow (-S^x_{2n}, -S^z_{2n}, -S^y_{2n}).
\end{aligned} \tag{62}$$

$H_{\Gamma_1}$ in Eq. (61) can be mapped to the following 1D Kitaev model via the identification $\Gamma_1 = K$,

$$H_K = \sum_{\langle ij \rangle \in \gamma \text{ bond}} K S^\gamma_i S^\gamma_j, \tag{63}$$

in which the bond pattern for $\gamma$ is shown in Fig. 1 (a). On the other hand, it is known that the 1D spin-1/2 Kitaev model can be solved by a Jordan-Wigner transformation [45], whose spectrum contains a Majorana flat band and a helical Majorana. Therefore, $H_{\Gamma_1}$ is also exactly solvable with an infinite ground state degeneracy.

From this discussion, we see that the physics at $(\Gamma_1 = 0, \Gamma_2)$ and $(\Gamma_1, \Gamma_2 = 0)$ is different from the phase of emergent $SU(2)_1$ conformal invariance. Hence we expect that the spin-1/2 asymmetric gamma model has an emergent $SU(2)_1$ conformal symmetry in a neighborhood of $\Gamma_1 = \Gamma_2$, i.e., $\theta = \pi/4$. However, the $SU(2)_1$ conformal symmetry does not extend to the special points $\Gamma_1 = 0$ or $\Gamma_2 = 0$, indicating a phase transition in between. This is verified by our DMRG numerical simulations to be discussed shortly.

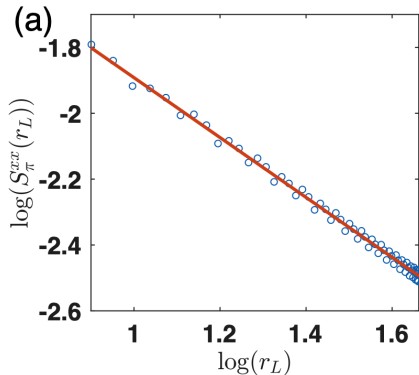
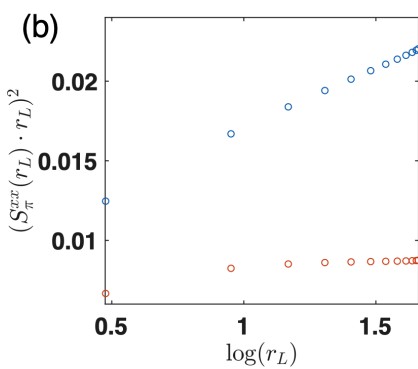

Figure 8: (a) $S_\pi^{xx}(r_L)$ as function of $r_L$ on a log-log scale where the slope is $-0.9089$, (b) $[r_L S_\pi^{xx}(r_L)]^2$ versus $\log(r_L)$ for $J_2 = 0$ (red points) and $J_2 = 0.075$ (black points), in which $r_L = \frac{L}{\pi}\sin(\frac{\pi r}{L})$. DMRG numerics are performed for the asymmetric gamma model defined in Eq. (55) at $\theta = 0.35\pi$ on a system of $L = 144$ sites using periodic boundary conditions.

## 4.3 Numerical evidence for emergent SU(2)$_1$ invariance

Next we discuss the numerical evidence for the emergent SU(2)$_1$ conformal invariance in the asymmetric gamma model. We compare the numerical results on central charge and spin correlation functions with the predictions from the SU(2)$_1$ WZW model. Because of the equivalent relations in Eqs. (58,59,60), we restrict to the range $\theta \in [\pi/4, \pi/2]$. Notice that $\theta = \pi/4$ is the symmetric gamma model, and $\theta = \pi/2$ corresponds to $\Gamma_1 = 0$.

According to Ref. [24], the $\theta = \pi/4$ point (i.e., the spin-1/2 1D symmetric gamma model) has an emergent SU(2)$_1$ invariance at low energies. Based on the analysis in Sec. 3.2, we expect that there is a range of $\theta$ around $\theta = \pi/4$ which has emergent SU(2)$_1$ invariance. Fig. 8 (a) shows the numerical results at $\theta = 0.35\pi$ for $S_\pi^{xx}(r_L)$ as a function of $r_L$ on a log-log scale, obtained from DMRG simulations on a system of $L = 144$ sites using periodic boundary conditions. The extracted exponent from the slope of the fitted line in Fig. 8 (a) is $-0.9089$, very close to the predicted value $-1$ in Eq. (43). The 9% deviation from 1 arises from the logarithmic correction in Eq. (43). To further study the logarithmic correction, we plot $[S_\pi^{xx}(r_L)r_L]^2$ as a function of $\log(r_L)$ as shown by the red dots in Fig. 8 (b). It is clear that the red dots approximately have a linear relation, which is consistent with the prediction in Eq. (43). Furthermore, the logarithmic correction in $S_\pi^{xx}$ can be killed by introducing a second nearest neighbor Heisenberg term $J_2 \sum_i \vec{S}_i \cdot \vec{S}_{i+2}$ into $H'_{A\Gamma}$ in Eq. (55) [40], similar to Sec. ??. As shown by the black dots in Fig. 8 (b), the relation between $[S_\pi^{xx}(r_L)r_L]^2$ and $\log(r_L)$ has already become very flat at $J_2 = 0.075$, indicating a significant suppression of the logarithmic correction and a critical value $J_{2c}$ very close to 0.075.

To determine the range of the phase of emergent SU(2)$_1$ conformal invariance, we have numerically calculated the central charge (denoted as $c$) in the narrow region $\theta \in [0.45\pi, 0.5\pi]$, as shown in Fig. 9 (a). Clearly, the value of the central charge remains very close to 1 until $\theta = 0.49\pi$, where it suddenly drops to zero, indicating an SU(2)$_1$ phase in the region $\theta \in [0.25\pi, 0.49\pi]$ and a different phase for $\theta \in [0.49\pi, 0.5\pi]$. In Fig. 9 (b), the fits for central charge at $\theta = 0.4625\pi$ and $\theta = 0.49\pi$ are shown by the black and red lines, respectively. It can be seen from Fig. 9 (b) that a good linear fit with $c = 0.946$ can be obtained from the black points, whereas the red points are far from a $c \sim 1$ linear relation. As discussed in Sec. 4.2, the existence of a phase different from SU(2)$_1$ in the neighborhood of $\theta = 0.5\pi$ is expected, since $\theta = 0.5\pi$ is an exactly solvable point which has an infinite ground state degeneracy.

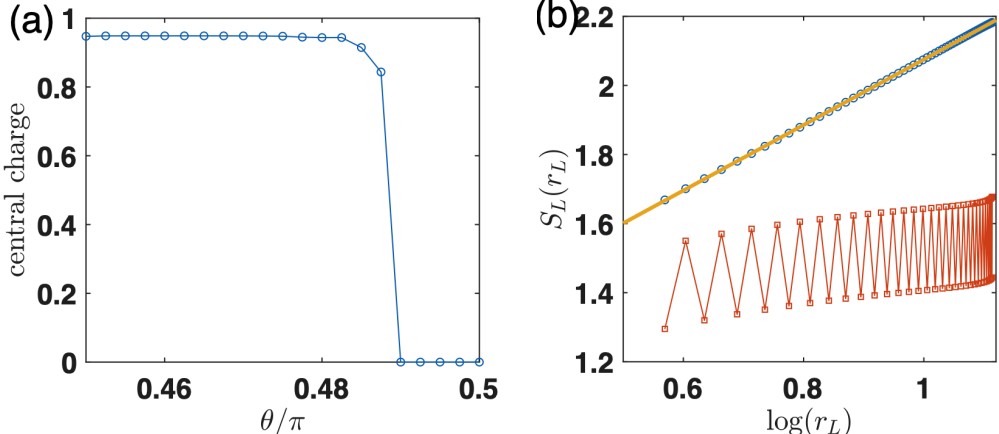

Figure 9: (a) Extracted values of the central charge for the asymmetric gamma model in the region $\theta \in [0.45\pi, 0.5\pi]$, (b) numerical data for the linear fits of the central charge at $\theta = 0.4625\pi$ (black) and $\theta = 0.49\pi$ (red). DMRG numerics are performed for the asymmetric gamma model defined in Eq. (55) on a system of $L = 144$ sites using periodic boundary conditions.

## 5 Nonsymmorphic cubic $O$ group

In this section, we discuss the nonsymmorphic cubic $O$ group, construct the minimal model having emergent $SU(2)_1$ conformal invariance, and present numerical evidence obtained from DMRG simulations.

### 5.1 Construction of the nonsymmorphic $O$ group

The cubic $O$ group contains 24 group elements. In the language of crystalline point groups, the $O$ group can be obtained from the $O_h$ group by removing the spatial inversion operation, i.e., $O_h \cong O \times \mathbb{Z}_2$. In our case, time reversal acts as the inversion in the spin space, hence the nonsymmorphic $O$ group $G_O$ can be constructed as

$$G_O = <R_a T_a, R_I I, R(\hat{x}, \pi), R(\hat{y}, \pi), R(\hat{z}, \pi)>,\qquad(64)$$

which satisfies

$$G_O/<T_{3a}> \cong O.\qquad(65)$$

It is useful and interesting to construct the generators of $G_O$. Let's define

$$\begin{aligned}R' &= (R_a T_a)^{-1},\\S' &= (R_a T_a)^{-1} \cdot R_I I \cdot R_a T_a \cdot R(\hat{y}, \pi).\end{aligned}\qquad(66)$$

It can be verified that the actions of $R'$ and $S'$ are given by

$$\begin{aligned}R' &: \quad (S_i'^{x}, S_i'^{y}, S_i'^{z}) \to (S_{i-1}'^{y}, S_{i-1}'^{z}, S_{i-1}'^{x}),\\S' &: \quad (S_i'^{x}, S_i'^{y}, S_i'^{z}) \to (S_{2-i}'^{y}, -S_{2-i}'^{x}, S_{2-i}'^{z}).\end{aligned}\qquad(67)$$

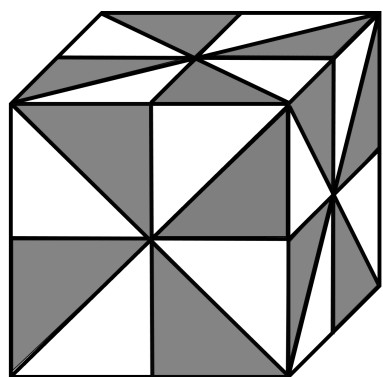

Figure 10: Decorated cube with the symmetry group as the $O$ group.

In fact, the group $G_O$ can be generated by $R'$, $S'$, i.e.,

$$G_O = <R', S'>, \tag{68}$$

which can be seen from the following constructions by comparing with Eq. (64),

$$
\begin{aligned}
R_a T_a &= (R')^{-1}, \\
R_I I &= (S')^2 R'(S')^{-1}(R')^3, \\
R(\hat{x}, \pi) &= R'(S')^2(R')^{-1}, \\
R(\hat{y}, \pi) &= (R')^{-1}(S')^2 R', \\
R(\hat{z}, \pi) &= (S')^2.
\end{aligned}
\tag{69}
$$

The cubic point group $O$ is isomorphic to the permutation group $S_4$, which has a generator-relation representation

$$O = <R, S | R^3 = S^4 = (RS)^2 = e>. \tag{70}$$

It can be verified that

$$
\begin{aligned}
(R')^3 &= T_{-3a}, \\
(S')^4 &= 1, \\
(R'S')^2 &= 1,
\end{aligned}
\tag{71}
$$

which satisfy the relations in Eq. (70) in the sense of modulo $T_{3a}$. Therefore, $<R', S'>/<T_{3a}> \subseteq O$. In addition, it can be verified that $<R', S'>/<T_{3a}>$ contains at least 24 distinct group elements. Since $|O| = 24$, we conclude that

$$<R', S'>/<T_{3a}> \cong O, \tag{72}$$

which proves Eq. (65).

We note that there is an intuitive understanding of the isomorphism in Eq. (65). If the spatial components in the symmetry operations in Eq. (64) are temporarily ignored, then it can be observed that all the operations restricted within the spin space leave the decorated cube in Fig. 10 invariant. On the other hand, the symmetry group of the decorated cube in Fig. 10 is the cubic $O$ group, hence it is not a surprise that $G_O$ is intimately related to the cubic $O$ group. Notice that Fig. 10 has a larger symmetry group than Fig. 3, since Fig. 10 is also invariant under $R_I$ shown in Fig. 2, while Fig. 3 is not.

Next, we derive the nonsymmorphic nonabelian bosonization formulas which are consistent with the nonsymmorphic $O$ group. Since $T$ is a subgroup of $O$, the relations in Eqs. (38,39) also apply to the $O$ case. Further imposing the $R_I I$ symmetry, we obtain the following constraints for the bosonization coefficients

$$
\begin{aligned}
D_2^{(L)} &= D_3^{(L)} , \\
D_2^{(R)} &= D_3^{(R)} , \\
C_2 &= C_3 ,
\end{aligned}
\tag{73}
$$

in addition to those in Eqs. (38,39). Using the nonsymmorphic bosonization formulas and the $SU(2)_1$ conformal field theory, the component $S_\pi^{\alpha\alpha}(r)$ of the spin correlation function $\langle S_1^\alpha S_{1+r}^\alpha \rangle$ ($\alpha = x, y, z$) in the $U_6$ frame can be shown to be given by Eq. (43) in which $C_2$ and $C_3$ should be set as equal.

## 5.2 The gamma-octupole model

We consider the following "gamma-octupole model"

$$
H_{\Gamma\Omega} = \Gamma \sum_{<ij>\in\gamma\,\text{bond}} \left( S_i^\alpha S_j^\beta + S_i^\beta S_j^\alpha \right) + \Omega \sum_j \left( S_{j-1}^x S_j^y S_{j+1}^x - S_{j-1}^y S_j^x S_{j+1}^y \right) ,
\tag{74}
$$

which in addition to $H_{S\Gamma}$, also contains a spin-octupolar term (i.e., the $\Omega$ term).

Performing the six-sublattice rotation $U_6$ defined in Eq. (2), $H_{\Gamma\Omega}$ becomes $H'_{\Gamma\Omega} = (U_6)^{-1} H_{\Gamma\Omega} U_6$, given by

$$
H'_{\Gamma\Omega} = -\Gamma \sum_{<ij>\in\gamma\,\text{bond}} \left( S_i'^\alpha S_j'^\alpha + S_i'^\beta S_j'^\beta \right) + \Omega \sum_j \left( S_{j-1}'^{\alpha_l} S_j'^{\gamma_l} S_{j+1}'^{\beta_l} - S_{j-1}'^{\alpha_r} S_j'^{\gamma_r} S_{j+1}'^{\beta_r} \right) ,
\tag{75}
$$

in which $\gamma = x, z, y$ has a three-site periodicity as shown in Fig. 1 (b); $\gamma_l = <j-1, j>$ and $(\alpha_l, \beta_l, \gamma_l)$ form a right-handed coordinate system; $\gamma_r = <j, j+1>$ and $(\alpha_r, \beta_r, \gamma_r)$ form a right-handed coordinate system. Explicit expressions of $H_{\Gamma\Omega}$ and $H'_{\Gamma\Omega} = (U_6)^{-1} H_{\Gamma\Omega} U_6$ are included in Appendix A.

Because of the spin-octupolar term, it is clear that $H'_{\Gamma\Omega}$ does not have time reversal symmetry. However, as can be checked, $H'_{\Gamma\Omega}$ is invariant under all other symmetries in Eq. (4) except $\mathcal{T}$. Therefore, the symmetry group $G_{\Gamma\Omega}$ is

$$
G_{\Gamma\Omega} = G_O ,
\tag{76}
$$

where $G_O$ is defined in Eq. (64). This shows that $H_{\Gamma\Omega}$ provides a concrete realization for the nonsymmorphic $O$ group, and it is expected that the system has an emergent $SU(2)_1$ conformal symmetry at low energies for a range of $\Omega$ around zero. We note that there are many other terms which preserve the nonsymmorphic $O$ symmetry, and the choice of the $\Omega$-term is only one such possibility.

Next, we discuss numerical evidence for the emergent $SU(2)_1$ invariance by comparing numerical results with the prediction in Eq. (43). Fig. 11 shows the numerical results of $S_\pi^{xx}(r_L)$ as a function of $r_L$ on a log-log scale for the gamma-octupole model in the $U_6$ frame defined in Eq. (75) at $\Gamma = -1$, and $\Omega = 0.3$, obtained from DMRG simulations on a system of $L = 144$ sites using periodic boundary conditions. The slope extracted from Fig. 4 (a) is $-1.043$, which is very close to $-1$, consistent with the prediction of $SU(2)_1$ WZW model in Eq. (43).

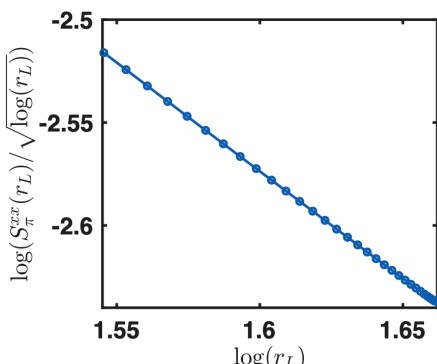

Figure 11: $S_\pi^{xx}(r_L)/\ln^{1/2}(r_L)$ as function of $r_L$ on a log-log scale where the slope is $-1.043$, in which $r_L = \frac{L}{\pi}\sin(\frac{\pi r}{L})$. DMRG numerics are performed for the gamma-octupole model in the $U_6$ frame on a system of $L = 144$ sites using periodic boundary conditions. The parameters are chosen as $\Gamma = -1$, $\Omega = 0.3$.

## 6 Nonsymmorphic cubic $T_d$ group

In this section, we construct the nonsymmorphic cubic $T_d$ group, give the minimal model for $T_d$ group which has emergent $\text{SU}(2)_1$ conformal invariance, and present numerical evidence obtained from DMRG simulations.

### 6.1 Construction of the nonsymmorphic $T_d$ group

The cubic $T_d$ group contains 24 group elements. The $T_d$ group is isomorphic to $S_4$, similar to the $O$ group. However, the difference is that while all the elements in $O$ are proper (i.e., having determinant equal to 1), $T_d$ contain 12 improper elements (with determinant equal to $-1$).

The generators of the nonsymmorphic cubic $T_d$ group $G_{T_d}$ can be obtained by slightly modifying the generators for $G_O$. Adding $\mathcal{T}$ to $S'$ in Eq. (66), we define

$$R'' = (R_a T_a)^{-1},$$
$$S'' = \mathcal{T} \cdot (R_a T_a)^{-1} \cdot R_I I \cdot R_a T_a \cdot R(\hat{y}, \pi). \tag{77}$$

The group $G_{T_d}$ is generated by the two generators in Eq. (77), i.e.,

$$G_{T_d} = <R'', S''>. \tag{78}$$

Using the same method in Sec. 5.1, it can be straightforwardly seen that $G_{T_d}$ satisfies

$$G_{T_d}/<T_{3a}> \cong S_4. \tag{79}$$

The difference from the nonsymmorphic cubic group $G_O$ lies in the additional $\mathcal{T}$ operation in the definition of $S''$, which generates improper symmetry operations in $G_{T_d}$.

We note that there is an intuitive understanding of the isomorphism in Eq. (79). If the spatial components in the symmetry operations in Eq. (77) are temporarily ignored, then it can be observed that all the operations restricted within the spin space leave the decorated cube in Fig. 12 invariant. On the other hand, the symmetry group of the decorated cube in Fig. 12 is the cubic $T_d$ group, hence it is not a surprise that $G_{T_d}$ is intimately related to the cubic $T_d$ group. Notice that Fig. 12 has a larger symmetry group than Fig. 3, since the former is also invariant under $R_I$ followed by an inversion (corresponding to time reversal) in Fig. 2, while the latter is not.

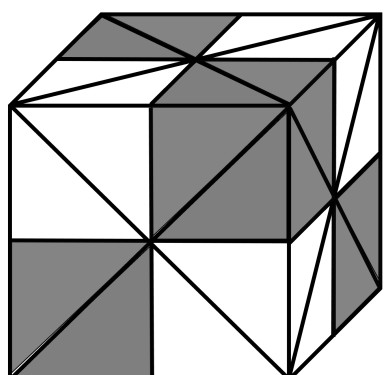

Figure 12: Decorated cube with the symmetry group as the $T_d$ group.

Next, we derive the nonsymmorphic nonabelian bosonization formulas which are consistent with the nonsymmorphic $T_d$ group. Since $T$ is a subgroup of $T_d$, the relations in Eqs. (38,39) also apply to the $T_d$ case. Further imposing the $\mathcal{T}R_I I$ symmetry, we obtain the following constraints for the bosonization coefficients

$$
\begin{aligned}
D_2^{(L)} &= D_3^{(R)}, \\
D_2^{(R)} &= D_3^{(L)}, \\
C_2 &= C_3,
\end{aligned}
\tag{80}
$$

in addition to those in Eqs. (38,39). Using the nonsymmorphic bosonization formulas and the $\mathrm{SU}(2)_1$ conformal field theory, the component $S_\pi^{\alpha\alpha}(r)$ of the spin correlation function $\langle S_1^\alpha S_{1+r}^\alpha \rangle$ ($\alpha = x, y, z$) in the $U_6$ frame can be shown to be given by Eq. (43) in which $C_2$ and $C_3$ should be set as equal.

## 6.2 The gamma-staggered-octupole model

We consider the following "gamma-staggered-octupole model"

$$
H_{\Gamma\Omega_2} = H_{S\Gamma} + \Omega_2 \sum_j (-)^{j-1} \left( S_{j-1}^x S_j^y S_{j+1}^x + S_{j-1}^y S_j^x S_{j+1}^y \right),
\tag{81}
$$

where $H_{S\Gamma}$ is defined in Eq. (1), and the $\Omega_2$ term represents a spin-octupolar interaction with a staggered sign.

Performing the six-sublattice rotation $U_6$ defined in Eq. (2), $H_{\Gamma\Omega_2}$ becomes $H'_{\Gamma\Omega_2} = (U_6)^{-1} H_{\Gamma\Omega_2} U_6$, given by

$$
H'_{\Gamma\Omega_2} = -\Gamma \sum_{<ij> \in \gamma \, \mathrm{bond}} \left( S_i'^{\alpha} S_j'^{\alpha} + S_i'^{\beta} S_j'^{\beta} \right) + \Omega_2 \sum_j \left( S_{j-1}'^{\alpha_l} S_j'^{\gamma_l} S_{j+1}'^{\beta_l} + S_{j-1}'^{\alpha_r} S_j'^{\gamma_r} S_{j+1}'^{\beta_r} \right),
\tag{82}
$$

in which $\gamma = x, z, y$ has a three-site periodicity as shown in Fig. 1 (b); $\gamma_l = <j-1, j>$ and $(\alpha_l, \beta_l, \gamma_l)$ form a right-handed coordinate system; $\gamma_r = <j, j+1>$ and $(\alpha_r, \beta_r, \gamma_r)$ form a right-handed coordinate system. Explicit expressions of $H_{\Gamma\Omega_2}$ and $H'_{\Gamma\Omega_2}$ are included in Appendix A.

It can be checked that $H'_{\Gamma\Omega_2}$ is not invariant under $\mathcal{T}$ nor $R_I I$, but the combination $\mathcal{T}R_I I$ is a symmetry of $H'_{\Gamma\Omega_2}$. Furthermore, it is straightforward to see that all the elements in $G_T$ are symmetries of $H'_{\Gamma\Omega_2}$, hence the symmetry group $G_{\Gamma\Omega_2}$ of $H'_{\Gamma\Omega_2}$ is

$$
G_{\Gamma\Omega_2} = <\mathcal{T}R_I I, R_a T_a, R(\hat{x}, \pi), R(\hat{y}, \pi), R(\hat{z}, \pi)>.
\tag{83}
$$

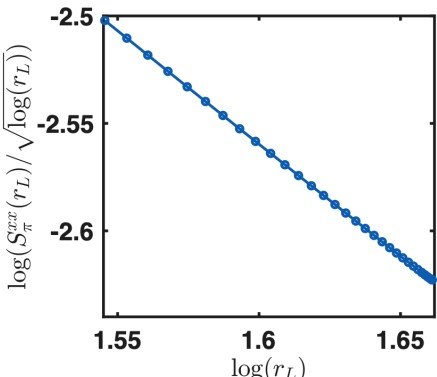

Figure 13: $S_\pi^{xx}(r_L)/\ln^{1/2}(r_L)$ as function of $r_L$ on a log-log scale where the slope is $-1.043$, in which $r_L = \frac{L}{\pi}\sin(\frac{\pi r}{L})$. DMRG numerics are performed for the gamma-staggered-octupole model in the $U_6$ frame on a system of $L = 144$ sites using periodic boundary conditions. The parameters are chosen as $\Gamma = -1$, $\Omega_2 = 0.3$.

It is not hard to show that

$$G_{\Gamma\Omega_2} = G_{T_d}, \tag{84}$$

where $G_{T_d}$ is defined in Eq. (78). To see this, simply notice that one the one hand, the generators $R''$ and $S''$ of $G_{T_d}$ can be obtained from the elements within the bracket of right hand side of Eq. (83); and on the other hand, all the elements in the bracket of right hand side of Eq. (83) can be constructed from $R''$ and $S''$, since

$$\mathcal{T}R_I I = (R'')^{-1}S''(R'')^{-1}(S'')^{-2}(R'')^2, \tag{85}$$

and the constructions for $R_a T_a$, $R(\hat{\alpha}, \pi)$ ($\alpha = x, y, z$) are the same as those in Eq. (69) except that $R'$ and $S'$ should be replaced by $R''$ and $S''$, respectively.

The above discussions show that $H_{\Gamma\Omega_2}$ provides a concrete realization for the nonsymmorphic $T_d$ group, and we expect that the system has an emergent $SU(2)_1$ conformal symmetry at low energies for a range of $\Omega_2$ around $\Omega_2 = 0$. We note that there are many other terms which preserve the nonsymmorphic $T_d$ symmetry, and the choice of the $\Omega_2$-term is only one such possibility.

Next, we discuss numerical evidence for the emergent $SU(2)_1$ invariance by comparing the numerical results with the predictions in Eq. (43). Fig. 13 shows the numerical results of $S_\pi^{xx}(r_L)/\ln^{1/2}(r_L)$ as a function of $r_L$ on a log-log scale for the gamma-staggered-octupole model in the $U_6$ frame defined in Eq. (82) at $\Gamma_1 = -1$ and $\Omega_2 = 0.3$, obtained from DMRG simulations on a system of $L = 144$ sites using periodic boundary conditions, in which $r_L = \frac{L}{\pi}\sin(\frac{\pi r}{L})$. The slope extracted from Fig. 13 (a) is $-1.043$, which is very close to $-1$, consistent with the prediction from $SU(2)_1$ WZW model in Eq. (43).

## 7 The cases of six-site unit cells

In previous sections, the systems all have a three-site periodicity in the $U_6$ frame. Remarkably, as shown in Ref. [34], it is possible for an emergence of $SU(2)_1$ conformal symmetry even when the unit cell contains six sites. This is a surprising result since naively the six-site unit cell corresponds to an integer spin, not satisfying the conditions of the Lieb-Schultz-Mattis-Affleck theorem [46–49] where half-odd integer spin is required. It has been established in Ref. [34] that the $SU(2)_1$ invariance is protected by a nonsymmorphic symmetry group $G_{\Gamma D_M}$ satisfying $G_{\Gamma D_M}/<T_{6a}> \cong O_h$ in the $U_6$ frame.

In this section, we ask the similar questions as Sec. 2.3: Can we lower the nonsymmorphic $O_h$ symmetry for the case of six-site unit cell while maintaining the emergent $SU(2)_1$ conformal invariance at low energies; and what is the smallest nonsymmorphic symmetry group which can stabilize an extended $SU(2)_1$ phase in this case? For a brief summary of the results in this section, we find that not all nonsymmorphic cubic symmetry groups can stabilize an $SU(2)_1$ phase in the case of six-site unit cell. In fact, besides $O_h$, only the $O$ and $T_d$ groups can do the job.

## 7.1 Review of the symmetric 1D gamma mode with a Dzyaloshinskii-Moriya interation

In this subsection, we briefly review the 1D spin-1/2 symmetric gamma model with an additional Dzyaloshinskii-Moriya (DM) interaction studied in detail in Ref. [34]. By adding $D_M(S_i^\alpha S_j^\beta - S_i^\beta S_j^\alpha)$ to the Hamiltonian on bond $\gamma = <ij>$ in Eq. (1), we obtain the following Kitaev-DM model,

$$H_{\Gamma D_M} = \sum_{<ij>\in\gamma\,\text{bond}} \left(\Gamma_1 S_i^\alpha S_j^\beta + \Gamma_2 S_i^\beta S_j^\alpha\right), \tag{86}$$

where $\Gamma_1 = \Gamma + D_M$, $\Gamma_2 = \Gamma - D_M$. After performing the $U_6$ transformation, the transformed Hamiltonian $H'_{\Gamma D_M} = (U_6)^{-1} H_{\Gamma D_M} U_6$ becomes

$$H'_{\Gamma D_M} = \sum_{<ij>\in\gamma\,\text{bond}} \left(-\Gamma_1 S_i^\alpha S_j^\alpha - \Gamma_2 S_i^\beta S_j^\beta\right), \tag{87}$$

in which this time, the bond $\gamma$ has a six-site periodicity as shown in Fig. 1 (c), and the conventions for the spin directions in Eq. (87) are: $(\gamma,\alpha,\beta)$ form a right-handed coordinate system when $\gamma \in \{x,y,z\}$; $(\gamma,\alpha,\beta)$ form a left-handed coordinate system when $\gamma \in \{\bar{x},\bar{y},\bar{z}\}$; and $S_j^\mu = S_j^{\bar{\mu}}$ ($\mu = x,y,z$). Explicit expressions of $H_{\Gamma D_M}$ and $H'_{\Gamma D_M}$ are included in Appendix A.2.1.

In the $U_6$ frame, $H'_{\Gamma D_M}$ is invariant under the following transformations

$$
\begin{aligned}
&1. \quad \mathcal{T} &&: \quad (S_i'^x, S_i'^y, S_i'^z) \to (-S_i'^x, -S_i'^y, -S_i'^z),\\
&2. \quad R_a^{-1} T_{2a} &&: \quad (S_i'^x, S_i'^y, S_i'^z) \to (S_{i+2}'^y, S_{i+2}'^z, S_{i+2}'^x),\\
&3. \quad R_I I &&: \quad (S_i'^x, S_i'^y, S_i'^z) \to (-S_{4-i}'^z, -S_{4-i}'^y, -S_{4-i}'^x),\\
&4. \quad R(\hat{x},\pi) &&: \quad (S_i'^x, S_i'^y, S_i'^z) \to (S_i'^x, -S_i'^y, -S_i'^z),\\
&5. \quad R(\hat{y},\pi) &&: \quad (S_i'^x, S_i'^y, S_i'^z) \to (-S_i'^x, S_i'^y, -S_i'^z),\\
&6. \quad R(\hat{z},\pi) &&: \quad (S_i'^x, S_i'^y, S_i'^z) \to (-S_i'^x, -S_i'^y, S_i'^z).
\end{aligned}
\tag{88}
$$

Clearly, though $H'_{\Gamma D_M}$ is invariant under $T_{6a}$, $T_{3a}$ is no longer a symmetry of the model. Comparing with Eq. (4), it can be seen that the only difference is a replacement of $R_a T_a$ by $(R_a T_a)^2 = R_a^{-1} T_{2a}$. The symmetry group $G_{\Gamma D_M}$ for the gamma-DM model in the $U_6$ frame is generated by the symmetry operations in Eq. (88), and it has been proved in Ref. [34] that $G_{\Gamma D_M}$ satisfies

$$G_{\Gamma D_M}/<T_{6a}> \cong O_h. \tag{89}$$

Since Eq. (5) for the symmetric gamma model can be alternatively rewritten as $G_{S\Gamma}/<T_{3a}> \cong O_h \times \mathbb{Z}_2$ where $\mathbb{Z}_2 = <T_{3a}>/<T_{6a}>$, we see that $G_{\Gamma D_M}$ is halved compared with $G_{S\Gamma}$.

As discussed in Ref. [34], in the phase diagram parametrized by $\theta$ (where $\theta$ is defined through $\Gamma_1 = \cos(\theta)$, $\Gamma_2 = \sin(\theta)$), $H'_{\Gamma D_M}$ has an extended gapless phase having emergent $SU(2)_1$ conformal invariance at low energies. The nonsymmorphic nonabelian bosonization formulas are given by

$$S^\alpha_{j+6n} = D^\alpha_{L,j} J^\alpha_L + D^\alpha_{R,j} J^\alpha_R + (-)^j C^\alpha_j N^\alpha \,, \tag{90}$$

in which the bosonization coefficients satisfy ($\nu = L, R$)

$$
\begin{aligned}
D^x_{\nu,1} &= D^y_{\nu,3} = D^z_{\nu,5} = D^y_{\nu,1} = D^z_{\nu,3} = D^x_{\nu,5} = D_2 \,, \\
D^z_{\nu,1} &= D^x_{\nu,3} = D^y_{\nu,5} = D_1 \,, \\
D^x_{\nu,2} &= D^y_{\nu,4} = D^z_{\nu,6} = D'_2 = D^z_{\nu,2} = D^x_{\nu,4} = D^y_{\nu,6} = D'_2 \,, \\
D^y_{\nu,2} &= D^z_{\nu,4} = D^x_{\nu,6} = D'_1 \,.
\end{aligned}
\tag{91}
$$

## 7.2 Symmetry analysis of the low energy field theory

In this subsection, we perform a symmetry analysis of the low energy field theory to figure out what nonsymmorphic symmetry groups can stabilize a gapless of emergent $SU(2)_1$ conformal invariance at low energies. The 1D spin-1/2 gamma-DM model in Eq. (87) is taken as the unperturbed starting point for the analysis.

Using the methods similar to Sec. 3.1, Sec. 4.1, Sec. 5.1, Sec. 6.1, the nonsymmorphic cubic groups $T$, $T_h$, $O$ and $T_d$ in the present case of six-site unit cells can be constructed as

$$\tilde{G}_T = <R_a^{-1} T_{2a}, R(\hat{x}, \pi), R(\hat{y}, \pi), R(\hat{z}, \pi)> \,, \tag{92}$$

$$\tilde{G}_{T_h} = <\mathcal{T}, R_a^{-1} T_{2a}, R(\hat{x}, \pi), R(\hat{y}, \pi), R(\hat{z}, \pi)> \,, \tag{93}$$

$$\tilde{G}_O = <R_I I, R_a^{-1} T_{2a}, R(\hat{x}, \pi), R(\hat{y}, \pi), R(\hat{z}, \pi)> \,, \tag{94}$$

$$\tilde{G}_{T_d} = <\mathcal{T} R_I I, R_a^{-1} T_{2a}, R(\hat{x}, \pi), R(\hat{y}, \pi), R(\hat{z}, \pi)> \,. \tag{95}$$

We note that neither $\tilde{G}_T$ nor $\tilde{G}_{T_h}$ can stabilize a gapless phase with emergent $SU(2)_1$ invariance, since $\epsilon = \text{tr}(g)$ is allowed by both symmetry groups, which is a relevant operator in the RG sense and opens a gap in the system.

Next we show that the low energy field theory remains to be the $SU(2)_1$ WZW model (up to marginally irrelevant operators) for both the $O$ and $T_d$ groups. The symmetry analysis is as follows.

1) Dimension 1/2 operators: $\epsilon = \text{tr}(g)$ is forbidden by $R_I I$ for $\tilde{G}_O$, and forbidden by $\mathcal{T} R_I I$ for $\tilde{G}_{T_d}$; $N^\alpha$ ($\alpha = x, y, z$) are forbidden by $R(\hat{\beta}, \pi)$ ($\beta \in \{x, y, z\}$, $\beta \neq \alpha$) for both $\tilde{G}_O$ and $\tilde{G}_{T_d}$.

2) Dimension 1 operators: $J^\alpha_\nu$ ($\alpha = x, y, z$ and $\nu = L, R$) are forbidden by $R(\hat{\beta}, \pi)$ ($\beta \neq \alpha$) for both $\tilde{G}_O$ and $\tilde{G}_{T_d}$.

3) Dimension 3/2 operators: $J^\alpha_L \epsilon$, $J^\alpha_R \epsilon$ are forbidden by $R(\hat{\beta}, \pi)$ ($\beta \neq \alpha$) for both $\tilde{G}_O$ and $\tilde{G}_{T_d}$; $(\vec{J}_L + \vec{J}_R) \cdot \vec{N}$ is allowed by $\tilde{G}_O$, whereas $\vec{J}_L \cdot \vec{N}$ and $\vec{J}_R \cdot \vec{N}$ are both allowed by $\tilde{G}_{T_d}$.

4) Dimension 2 operators: $\vec{J}_L \cdot \vec{J}_L + \vec{J}_R \cdot \vec{J}_R$ and $\vec{J}_L \cdot \vec{J}_R$ are allowed by $\tilde{G}_O$, whereas $\vec{J}_L \cdot \vec{J}_L$, $\vec{J}_R \cdot \vec{J}_R$ and $\vec{J}_L \cdot \vec{J}_R$ are allowed by $\tilde{G}_{T_d}$.

Hence, the low energy field theory compatible with $\tilde{G}_O$ is

$$\tilde{H}_O = \int dx \frac{2\pi}{3} v(\vec{J}_L \cdot \vec{J}_L + \vec{J}_R \cdot \vec{J}_R) + w \int dx(\vec{J}_L + \vec{J}_R) \cdot \vec{N} - u \int dx \vec{J}_L \cdot \vec{J}_R \,, \tag{96}$$

and the field theory compatible with $\tilde{G}_{T_d}$ is

$$\tilde{H}_O = \int dx \frac{2\pi}{3} v(\lambda \vec{J}_L \cdot \vec{J}_L + \lambda^{-1} \vec{J}_R \cdot \vec{J}_R) + \int dx(w_L \vec{J}_L \cdot \vec{N} + w_R \vec{J}_R \cdot \vec{N}) - u \int dx \vec{J}_L \cdot \vec{J}_R \,, \tag{97}$$

in which $v$ is velocity, $\lambda$, $w$, $w_L$, $w_R$, $u$ are coupling constants. On the other hand, it has been shown in Ref. [34] that both $\vec{J}_L \cdot \vec{N}$ and $\vec{J}_R \cdot \vec{N}$ are total derivatives in the SU(2)$_1$ WZW model, given by

$$(\vec{J}_L \cdot \vec{N})(z, \bar{z}') = -3i\partial_z \epsilon(z, \bar{z}'),$$
$$(\vec{J}_R \cdot \vec{N})(z, \bar{z}') = 3i\partial_{\bar{z}'} \epsilon(z, \bar{z}'), \tag{98}$$

in which $z = \lambda\tau + ix$ and $\bar{z}' = \lambda^{-1}\tau - ix$ are the holomorphic and anti-holomorphic coordinates, respectively, where $\tau$ is the imaginary time, $x$ is spatial coordinate, and appearances of $\lambda$ and $\lambda^{-1}$ in the expressions of $z$ and $\bar{z}'$ are due to the fact that the velocities may not be the same for left and right movers as discussed in Sec. 3.2. As a result, $\vec{J}_L \cdot \vec{N}$ and $\vec{J}_R \cdot \vec{N}$ have no effects on the low energy properties, since their space-time integration vanish in the action in the path integral. Then according to the RG analysis in Sec. 3.2, we see that as long as $u > 0$ in Eqs. (96,96), the system has an emergent SU(2)$_1$ conformal invariance at low energies. This establishes the fact that both the nonsymmorphic $O$ and $T_d$ groups can stabilize an extended SU(2)$_1$ phase in the present case of six-site unit cells.

The nonsymmorphic bosonization formulas can be derived similarly as before. For the $O$ group, the coefficients in Eq. (9) satisfy

$$D_{v,1}^x = D_{v,3}^y = D_{v,5}^z = D_{v,1}^y = D_{v,3}^z = D_{v,5}^x = D_2^{(v)},$$
$$D_{v,1}^z = D_{v,3}^x = D_{v,5}^y = D_1^{(v)},$$
$$D_{v,2}^x = D_{v,4}^y = D_{v,6}^z = D_{v,2}^z = D_{v,4}^x = D_{v,6}^y = D_2'^{(v)},$$
$$D_{v,2}^y = D_{v,4}^z = D_{v,6}^x = D_1'^{(v)}, \tag{99}$$

whereas for the $T_d$ group, they satisfy

$$D_{L,1}^x = D_{L,3}^y = D_{L,5}^z = D_{R,1}^y = D_{R,3}^z = D_{R,5}^x = D_2^{(L)},$$
$$D_{R,1}^x = D_{R,3}^y = D_{R,5}^z = D_{L,1}^y = D_{L,3}^z = D_{L,5}^x = D_2^{(R)},$$
$$D_{L,1}^z = D_{L,3}^x = D_{L,5}^y = D_{R,1}^z = D_{R,3}^x = D_{R,5}^y = D_1,$$
$$D_{L,2}^x = D_{L,4}^y = D_{L,6}^z = D_{R,2}^z = D_{R,4}^x = D_{R,6}^y = D_2'^{(L)},$$
$$D_{R,2}^x = D_{R,4}^y = D_{R,6}^z = D_{L,2}^z = D_{L,4}^x = D_{L,6}^y = D_2'^{(R)},$$
$$D_{L,2}^y = D_{L,4}^z = D_{L,6}^x = D_{R,2}^y = D_{R,4}^z = D_{R,6}^x = D_1'. \tag{100}$$

## 7.3 $O$ group: The gamma-DM-octupole model

The minimal model for the nonsymmorphic $O$ group with six-site unit cells in the $U_6$ frame is the gamma-DM-octupole model defined in the original frame as follows,

$$H_{\Gamma D_M \Omega} = H_{\Gamma D_M} + \Omega\left(\sum_j S_{j-1}^x S_j^y S_{j+1}^x - \sum_j S_{j-1}^y S_j^x S_{j+1}^y\right), \tag{101}$$

in which $H_{\Gamma D_M}$ is defined in Eq. (86). Explicit expressions of the Hamiltonians in the original and $U_6$ frames are included in Appendix A.2.2.

Using the nonsymmorphic nonabelian bosonization formulas in Eq. (99), the $\pi$-wavevector component $S_\pi^{xx}(r)$ of the spin correlation function $\langle S_1^x S_r^x \rangle$ in the $U_6$ frame can be derived as

$$S_\pi^{xx}(r) = \frac{A_x}{r^2} + \frac{B_x \ln^{1/2}(r/r_0)}{r}, \tag{102}$$

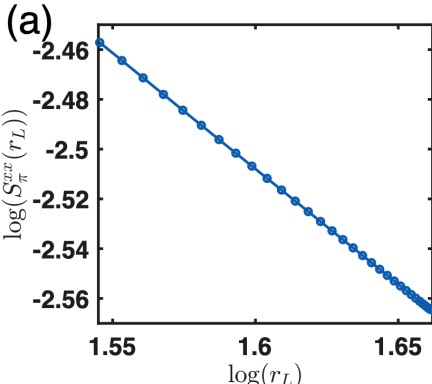 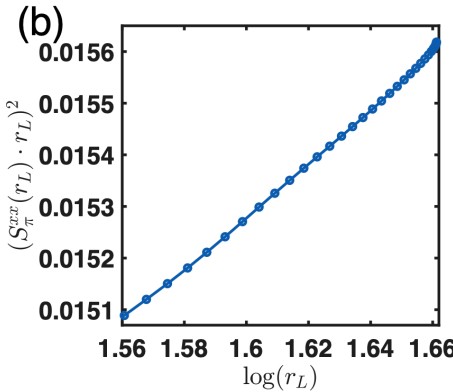

Figure 14: (a) $S_\pi^{xx}(r_L)$ as function of $r_L$ on a log-log scale where the slope is $-0.9266$, (b) $[r_L S_\pi^{xx}(r_L)]^2$ versus $\log(r_L)$, in which $r_L = \frac{L}{\pi}\sin(\frac{\pi r}{L})$. DMRG numerics are performed for the gamma-DM-octupole model in the $U_6$ frame on a system of $L = 144$ sites using periodic boundary conditions. The parameters are chosen as $\Gamma_1 = -0.8$, $\Gamma_2 = -0.12$, $\Omega = 0.3$.

in which

$$A_x = \frac{1}{6}\left[ -D_2^{(L)}\left(2D_2^{(L)} - 2D_2'^{(L)} + D_1^{(L)} - D_1'^{(L)}\right) + 2D_2^{(R)}\left(D_2^{(R)} + D_2'^{(R)} + D_1^{(R)} - D_1'^{(R)}\right)\right],$$
$$B_x = \frac{1}{6}C_2(2C_2 + 2C_2' + C_1 + C_1'). \tag{103}$$

Next, we discuss numerical evidence for the emergent $SU(2)_1$ invariance by comparing numerical results with the prediction in Eq. (102). In Fig. 14 (a), the numerical results of $S_\pi^{xx}(r_L)$ as a function of $r_L$ are shown for the gamma-DM-octupole model in the $U_6$ frame at $\Gamma_1 = -0.8$, $\Gamma_2 = -1.2$, and $\Omega = 0.3$, obtained from DMRG simulations on a system of $L = 144$ sites using periodic boundary conditions. In the $r \gg 1$ limit, the $1/r^2$ term in Eq. (102) can be ignored, hence Eq. (102) predicts an exponent close to 1. The slope extracted from Fig. 14 (a) is $-0.9266$, which is very close to $-1$, consistent with the prediction of $SU(2)_1$ WZW model. In fact, the 7% deviation of the exponent from 1 is due to the logarithmic correction in Eq. (102). To further study the logarithmic correction, $[r_L S_\pi^{xx}(r_L)]^2$ is plotted against $\log(r_L)$ as shown in Fig. 14 (b). As can be seen from Fig. 14 (b), the relation is very linear, consistent with the theoretical prediction in Eq. (102) in the $r \gg 1$ limit.

## 7.4 $T_d$ group: The gamma-DM-staggered-octupole model

The minimal model for the nonsymmorphic $T_d$ group with six-site unit cells in the $U_6$ frame is the gamma-DM-staggered-octupole model defined in the original frame as follows,

$$H_{\Gamma D_M \Omega_2} = H_{\Gamma D_M} + \Omega_2 \sum_j (-)^{j-1}(S_{j-1}^x S_j^y S_{j+1}^x + S_{j-1}^y S_j^x S_{j+1}^y), \tag{104}$$

in which $H_{\Gamma D_M}$ is defined in Eq. (86). Explicit expressions of the Hamiltonians in the original and $U_6$ frames are included in Appendix A.2.3.

Using the bosonization formulas in Eq. (100), the $\pi$-wavevector component $S_\pi^{xx}(r)$ of the spin correlation function $\langle S_1^x S_r^x \rangle$ in the $U_6$ frame can be derived as

$$S_\pi^{xx}(r) = \frac{A_x}{r^2} + \frac{B_x \ln^{1/2}(r/r_0)}{r}, \tag{105}$$

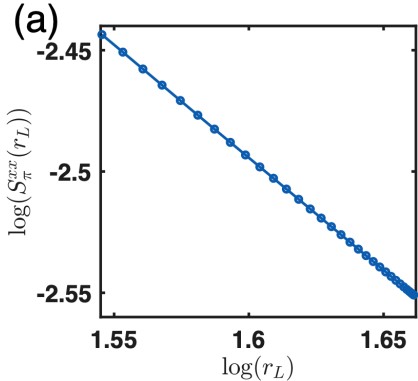 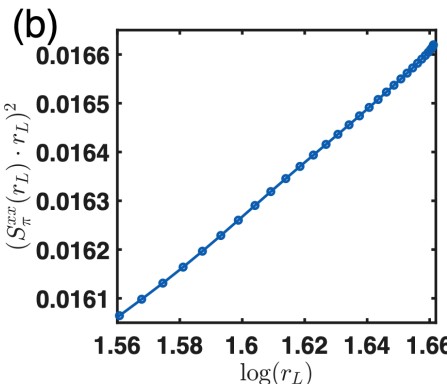

Figure 15: (a) $S_\pi^{xx}(r_L)$ as function of $r_L$ on a log-log scale where the slope is $-0.9272$, (b) $[r_L S_\pi^{xx}(r_L)]^2$ versus $\log(r_L)$, in which $r_L = \frac{L}{\pi}\sin(\frac{\pi r}{L})$. DMRG numerics are performed for the gamma-DM-staggered-octupole model in the $U_6$ frame on a system of $L = 144$ sites using periodic boundary conditions. The parameters are chosen as $\Gamma_1 = -0.8$, $\Gamma_2 = -0.12$, $\Omega_2 = 0.3$.

in which

$$A_x = \frac{1}{6}\Big[D_2^{(L)}(-D_2^{(L)} + D_2'^{(L)} - D_1 + D_2'^{(R)} - D_2^{(R)} + D_1')$$
$$+ D_2^{(R)}(-D_2^{(R)} + D_2'^{(R)} - D_1 + D_2'^{(L)} - D_2^{(L)} + D_1')\Big],$$
$$B_x = \frac{1}{6}C_2(2C_2 + 2C_2' + C_1 + C_1'). \tag{106}$$

Next, we discuss numerical evidence for the emergent $SU(2)_1$ invariance by comparing numerical results with the prediction in Eq. (105). In Fig. 15 (a), numerical results of $S_\pi^{xx}(r_L)$ as a function of $r_L$ on a log-log scale are shown for the gamma-DM-staggered-octupole model in the $U_6$ frame at $\Gamma_1 = -0.8$, $\Gamma_2 = -1.2$, and $\Omega = 0.3$, obtained from DMRG simulations on a system of $L = 144$ sites using periodic boundary conditions. In the $r \gg 1$ limit, the $1/r^2$ term in Eq. (105) can be ignored, hence Eq. (105) predicts an exponent close to 1. The slope extracted from Fig. 15 (a) is $-0.9272$, which is very close to $-1$, consistent with the prediction of $SU(2)_1$ WZW model. In fact, the 7% deviation of the exponent from 1 is due to the logarithmic correction in Eq. (105). To further study the logarithmic correction, $[r_L S_\pi^{xx}(r_L)]^2$ is plotted against $\log(r_L)$ as shown in Fig. 15 (b). As can be seen from Fig. 15 (b), the relation is very linear, consistent with the theoretical prediction in Eq. (105) in the $r \gg 1$ limit.

# 8 Conclusion

In summary, we have studied the nonsymmorphic groups which can lead to extended gapless phases with emergent $SU(2)_1$ conformal invariances in one-dimensional spin-1/2 models. We find that all the five nonsymmorphic cubic groups including $O_h$, $O$, $T_h$, $T_d$ and $T$ can stabilize $SU(2)_1$ phases, whereas nonsymmorphic planar groups cannot. Minimal models are constructed for the corresponding nonsymmorphic cubic groups, and numerical evidence of emergent $SU(2)_1$ conformal invariance are provided in the constructed models. Our work is useful for understanding gapless phases in 1D spin systems having nonsymmorphic symmetries.

# Acknowledgments

**Funding information** W.Y. acknowledges the startup funding at Nankai University. W.Y. and I.A. acknowledge support from NSERC Discovery Grant 04033-2016. A.N. acknowledges computational resources and services provided by Compute Canada and Advanced Research Computing at the University of British Columbia. A.N. acknowledges support from the Max Planck-UBC-UTokyo Center for Quantum Materials and the Canada First Research Excellence Fund (CFREF) Quantum Materials and Future Technologies Program of the Stewart Blusson Quantum Matter Institute (SBQMI). C.X. is partially supported by Strategic Priority Research Program of CAS (No. XDB28000000).

# A Explicit forms of the Hamiltonians

In this appendix, we give the explicit forms of the Hamiltonians for the models.

## A.1 Three-site unit cell

### A.1.1 $O_h$ group: Symmetric gamma model

In the original frame, the Hamiltonian is

$$
\begin{aligned}
H_{2n+1,2n+2} &= \Gamma(S_{2n+1}^y S_{2n+2}^z + S_{2n+1}^z S_{2n+2}^y), \\
H_{2n+2,2n+3} &= \Gamma(S_{2n+2}^z S_{2n+3}^x + S_{2n+2}^x S_{2n+3}^z).
\end{aligned} \tag{A.1}
$$

In the $U_6$ frame, the Hamiltonian is

$$
\begin{aligned}
H'_{3n+1,3n+2} &= -\Gamma(S_{3n+1}^{\prime y} S_{3n+2}^{\prime y} + S_{3n+1}^{\prime z} S_{3n+2}^{\prime z}), \\
H'_{3n+2,3n+3} &= -\Gamma(S_{3n+2}^{\prime x} S_{3n+3}^{\prime x} + S_{3n+2}^{\prime y} S_{3n+3}^{\prime y}), \\
H'_{3n+3,3n+4} &= -\Gamma(S_{3n+3}^{\prime z} S_{3n+4}^{\prime z} + S_{3n+3}^{\prime x} S_{3n+4}^{\prime x}).
\end{aligned} \tag{A.2}
$$

### A.1.2 $T$ group: Asymmetric-gamma-omega model

In the original frame, the Hamiltonian is

$$
\begin{aligned}
H_{2n+1,2n+2} &= \Gamma_1 S_{2n+1}^y S_{2n+2}^z + \Gamma_2 S_{2n+1}^z S_{2n+2}^y + \Omega(S_{2n}^x S_{2n+1}^y S_{2n+2}^x - S_{2n}^y S_{2n+1}^x S_{2n+2}^y), \\
H_{2n+2,2n+3} &= \Gamma_1 S_{2n+1}^x S_{2n+2}^z + \Gamma_2 S_{2n+1}^z S_{2n+2}^x + \Omega(S_{2n}^x S_{2n+1}^y S_{2n+2}^x - S_{2n}^y S_{2n+1}^x S_{2n+2}^y).
\end{aligned} \tag{A.3}
$$

In the $U_6$ frame, the Hamiltonian is

$$
\begin{aligned}
H'_{3n+1,3n+2} &= -(\Gamma_1 S_{3n+1}^{\prime y} S_{3n+2}^{\prime y} + \Gamma_2 S_{3n+1}^{\prime z} S_{3n+2}^{\prime z}) + \Omega(S_{3n}^{\prime z} S_{3n+1}^{\prime y} S_{3n+2}^{\prime x} - S_{3n}^{\prime y} S_{3n+1}^{\prime x} S_{3n+2}^{\prime z}), \\
H'_{3n+2,3n+3} &= -(\Gamma_1 S_{3n+2}^{\prime x} S_{3n+3}^{\prime x} + \Gamma_2 S_{3n+2}^{\prime y} S_{3n+3}^{\prime y}) + \Omega(S_{3n+1}^{\prime y} S_{3n+2}^{\prime x} S_{3n+3}^{\prime z} - S_{3n+1}^{\prime x} S_{3n+2}^{\prime z} S_{3n+3}^{\prime y}), \\
H'_{3n+3,3n+4} &= -(\Gamma_1 S_{3n+3}^{\prime z} S_{3n+4}^{\prime z} + \Gamma_2 S_{3n+3}^{\prime x} S_{3n+4}^{\prime x}) + \Omega(S_{3n+2}^{\prime x} S_{3n+3}^{\prime z} S_{3n+4}^{\prime y} - S_{3n+2}^{\prime z} S_{3n+3}^{\prime y} S_{3n+4}^{\prime x}).
\end{aligned} \tag{A.4}
$$

### A.1.3 $T_h$ group: Asymmetric gamma model

In the original frame, the Hamiltonian is

$$
\begin{aligned}
H_{2n+1,2n+2} &= \Gamma_1 S_{2n+1}^y S_{2n+2}^z + \Gamma_2 S_{2n+1}^z S_{2n+2}^y, \\
H_{2n+2,2n+3} &= \Gamma_1 S_{2n+1}^x S_{2n+2}^z + \Gamma_2 S_{2n+1}^z S_{2n+2}^x.
\end{aligned} \tag{A.5}
$$

In the $U_6$ frame, the Hamiltonian is

$$
\begin{aligned}
H'_{3n+1,3n+2} &= -(\Gamma_1 S'^{y}_{3n+1} S'^{y}_{3n+2} + \Gamma_2 S'^{z}_{3n+1} S'^{z}_{3n+2}), \\
H'_{3n+2,3n+3} &= -(\Gamma_1 S'^{x}_{3n+2} S'^{x}_{3n+3} + \Gamma_2 S'^{y}_{3n+2} S'^{y}_{3n+3}), \\
H'_{3n+3,3n+4} &= -(\Gamma_1 S'^{z}_{3n+3} S'^{z}_{3n+4} + \Gamma_2 S'^{x}_{3n+3} S'^{x}_{3n+4}).
\end{aligned}
\tag{A.6}
$$

### A.1.4   $O$ group: Gamma-omega model

In the original frame, the Hamiltonian is

$$
\begin{aligned}
H_{2n+1,2n+2} &= \Gamma(S^{y}_{2n+1} S^{z}_{2n+2} + S^{z}_{2n+1} S^{y}_{2n+2}) + \Omega(S^{x}_{2n} S^{y}_{2n+1} S^{x}_{2n+2} - S^{y}_{2n} S^{x}_{2n+1} S^{y}_{2n+2}), \\
H_{2n+2,2n+3} &= \Gamma(S^{z}_{2n+2} S^{x}_{2n+3} + S^{x}_{2n+2} S^{z}_{2n+3}) + \Omega(S^{x}_{2n} S^{y}_{2n+1} S^{x}_{2n+2} - S^{y}_{2n} S^{x}_{2n+1} S^{y}_{2n+2}).
\end{aligned}
\tag{A.7}
$$

In the $U_6$ frame, the Hamiltonian is

$$
\begin{aligned}
H'_{3n+1,3n+2} &= -\Gamma(S'^{y}_{3n+1} S'^{y}_{3n+2} + S'^{z}_{3n+1} S'^{z}_{3n+2}) + \Omega(S'^{z}_{3n} S'^{y}_{3n+1} S'^{x}_{3n+2} - S'^{y}_{3n} S'^{x}_{3n+1} S'^{z}_{3n+2}), \\
H'_{3n+2,3n+3} &= -\Gamma(S'^{x}_{3n+2} S'^{x}_{3n+3} + S'^{y}_{3n+2} S'^{y}_{3n+3}) + \Omega(S'^{y}_{3n+1} S'^{x}_{3n+2} S'^{z}_{3n+3} - S'^{x}_{3n+1} S'^{z}_{3n+2} S'^{y}_{3n+3}), \\
H'_{3n+3,3n+4} &= -\Gamma(S'^{z}_{3n+3} S'^{z}_{3n+4} + S'^{x}_{3n+3} S'^{x}_{3n+4}) + \Omega(S'^{x}_{3n+2} S'^{z}_{3n+3} S'^{y}_{3n+4} - S'^{z}_{3n+2} S'^{y}_{3n+3} S'^{x}_{3n+4}).
\end{aligned}
\tag{A.8}
$$

### A.1.5   $T_d$ group: Gamma-$\Omega_2$ model

In the original frame, the Hamiltonian is

$$
\begin{aligned}
H_{2n+1,2n+2} &= \Gamma(S^{y}_{2n+1} S^{z}_{2n+2} + S^{z}_{2n+1} S^{y}_{2n+2}) + \Omega_2(S^{x}_{2n} S^{y}_{2n+1} S^{x}_{2n+2} + S^{y}_{2n} S^{x}_{2n+1} S^{y}_{2n+2}), \\
H_{2n+2,2n+3} &= \Gamma(S^{z}_{2n+2} S^{x}_{2n+3} + S^{x}_{2n+2} S^{z}_{2n+3}) - \Omega_2(S^{x}_{2n} S^{y}_{2n+1} S^{x}_{2n+2} + S^{y}_{2n} S^{x}_{2n+1} S^{y}_{2n+2}).
\end{aligned}
\tag{A.9}
$$

In the $U_6$ frame, the Hamiltonian is

$$
\begin{aligned}
H'_{3n+1,3n+2} &= -\Gamma(S'^{y}_{3n+1} S'^{y}_{3n+2} + S'^{z}_{3n+1} S'^{z}_{3n+2}) + \Omega_2(S'^{z}_{3n} S'^{y}_{3n+1} S'^{x}_{3n+2} + S'^{y}_{3n} S'^{x}_{3n+1} S'^{z}_{3n+2}), \\
H'_{3n+2,3n+3} &= -\Gamma(S'^{x}_{3n+2} S'^{x}_{3n+3} + S'^{y}_{3n+2} S'^{y}_{3n+3}) + \Omega_2(S'^{y}_{3n+1} S'^{x}_{3n+2} S'^{z}_{3n+3} + S'^{x}_{3n+1} S'^{z}_{3n+2} S'^{y}_{3n+3}), \\
H'_{3n+3,3n+4} &= -\Gamma(S'^{z}_{3n+3} S'^{z}_{3n+4} + S'^{x}_{3n+3} S'^{x}_{3n+4}) + \Omega_2(S'^{x}_{3n+2} S'^{z}_{3n+3} S'^{y}_{3n+4} + S'^{z}_{3n+2} S'^{y}_{3n+3} S'^{x}_{3n+4}).
\end{aligned}
\tag{A.10}
$$

## A.2   Six-site unit cell

### A.2.1   $O_h$ group: Gamma-DM model

In the original frame, the Hamiltonian is

$$
\begin{aligned}
H_{2n+1,2n+2} &= \Gamma_1 S^{y}_{2n+1} S^{z}_{2n+2} + \Gamma_2 S^{z}_{2n+1} S^{y}_{2n+2}, \\
H_{2n+2,2n+3} &= \Gamma_1 S^{z}_{2n+2} S^{x}_{2n+3} + \Gamma_2 S^{x}_{2n+2} S^{z}_{2n+3}.
\end{aligned}
\tag{A.11}
$$

In the $U_6$ frame, the Hamiltonian is

$$
\begin{aligned}
H'_{6n+1,6n+2} &= -\Gamma_1 S'^{y}_{1+6n} S'^{y}_{2+6n} - \Gamma_2 S'^{z}_{1+6n} S'^{z}_{2+6n}, \\
H'_{6n+2,6n+3} &= -\Gamma_1 S'^{y}_{2+6n} S'^{y}_{3+6n} - \Gamma_2 S'^{x}_{2+6n} S'^{x}_{3+6n}, \\
H'_{6n+3,6n+4} &= -\Gamma_1 S'^{z}_{3+6n} S'^{z}_{4+6n} - \Gamma_2 S'^{x}_{3+6n} S'^{x}_{4+6n}, \\
H'_{6n+4,6n+5} &= -\Gamma_1 S'^{z}_{4+6n} S'^{z}_{5+6n} - \Gamma_2 S'^{y}_{4+6n} S'^{y}_{5+6n}, \\
H'_{6n+5,6n+6} &= -\Gamma_1 S'^{x}_{5+6n} S'^{x}_{6+6n} - \Gamma_2 S'^{y}_{5+6n} S'^{y}_{6+6n}, \\
H'_{6n+6,6n+7} &= -\Gamma_1 S'^{x}_{6+6n} S'^{x}_{7+6n} - \Gamma_2 S'^{z}_{6+6n} S'^{z}_{7+6n}.
\end{aligned}
\tag{A.12}
$$

### A.2.2  *O* group: Gamma-DM-octupole model

In the original frame, the Hamiltonian can be obtained by adding the $\Omega$ term in Eq. (A.7) to Eq. (A.11). In the $U_6$ frame, the Hamiltonian can be obtained by adding the $\Omega$ term in Eq. (A.8) to Eq. (A.12).

### A.2.3  $T_d$ group: Gamma-DM-staggered-octupole model

In the original frame, the Hamiltonian can be obtained by adding the $\Omega_2$ term in Eq. (A.9) to Eq. (A.11). In the $U_6$ frame, the Hamiltonian can be obtained by adding the $\Omega_2$ term in Eq. (A.10) to Eq. (A.12).

## B  Transformation properties of the SU(2)$_1$ WZW field

The transformation laws of $g$ and $\vec{J}_L, \vec{J}_R$ under time reversal $T$, spatial translation $T_a$, inversion $I$ and spin rotation $R \in SU(2)$ are summarized as

$$
\begin{aligned}
T: \quad & \epsilon(x) \rightarrow \epsilon(x), & & \vec{N}(x) \rightarrow -\vec{N}(x), \\
& \vec{J}_L(x) \rightarrow -\vec{J}_R(x), & & \vec{J}_R(x) \rightarrow -\vec{J}_L(x),
\end{aligned}
\tag{B.1}
$$

$$
\begin{aligned}
T_a: \quad & \epsilon(x) \rightarrow -\epsilon(x), & & \vec{N}(x) \rightarrow -\vec{N}(x), \\
& \vec{J}_L(x) \rightarrow \vec{J}_L(x), & & \vec{J}_R(x) \rightarrow \vec{J}_R(x),
\end{aligned}
\tag{B.2}
$$

$$
\begin{aligned}
I: \quad & \epsilon(x) \rightarrow -\epsilon(-x), & & \vec{N}(x) \rightarrow \vec{N}(-x), \\
& \vec{J}_L(x) \rightarrow \vec{J}_R(-x), & & \vec{J}_R(x) \rightarrow \vec{J}_L(-x),
\end{aligned}
\tag{B.3}
$$

$$
\begin{aligned}
R: \quad & \epsilon(x) \rightarrow \epsilon(x), & & N^{\alpha}(x) \rightarrow R^{\alpha}_{\beta} N^{\beta}(x), \\
& J^{\alpha}_L(x) \rightarrow R^{\alpha}_{\beta} J^{\beta}_L(x), & & J^{\alpha}_R(x) \rightarrow R^{\alpha}_{\beta} J^{\beta}_R(x),
\end{aligned}
\tag{B.4}
$$

in which $x$ is the spatial coordinate; $R^{\alpha}_{\beta}$ ($\alpha, \beta = x, y, z$) is the matrix element of the $3 \times 3$ rotation matrix $R$; $\epsilon(x) = \mathrm{tr}\, g(x)$ is the dimer order parameter; and $\vec{N}(x) = i\mathrm{tr}(g(x)\vec{\sigma})$ is the Néel order parameter [40].

## C  Spin correlation functions for the case of nonsymmorphic cubic *T* symmetry

The expressions of all the Fourier components in Eq. (42) are

$$
\begin{aligned}
S_0^{xx}(r) &= -\frac{1}{r^2} \cdot \frac{1}{3} \left[ D_2^{(L)} \left( D_1^{(L)} + D_2^{(L)} + D_3^{(L)} \right) + D_2^{(R)} \left( D_1^{(R)} + D_2^{(R)} + D_3^{(R)} \right) \right], \\
S_{\pi}^{xx}(r) &= \frac{\ln^{1/2}(r/r_0)}{r} \cdot \frac{1}{3} C_2 (C_1 + C_2 + C_3), \\
S_{\pi/3,(1)}^{xx}(r) &= \frac{\ln^{1/2}(r/r_0)}{r} \cdot \frac{1}{\sqrt{3}} C_2 (C_1 - C_3), \\
S_{\pi/3,(2)}^{xx}(r) &= \frac{\ln^{1/2}(r/r_0)}{r} \cdot \frac{1}{3} C_2 (-C_1 + 2C_2 - C_3), \\
S_{2\pi/3,(1)}^{xx}(r) &= -\frac{1}{r^2} \cdot \frac{1}{\sqrt{3}} \left[ D_2^{(L)} (D_3^{(L)} - D_1^{(L)}) + D_2^{(R)} (D_3^{(R)} - D_1^{(R)}) \right], \\
S_{2\pi/3,(2)}^{xx}(r) &= -\frac{1}{r^2} \cdot \frac{1}{3} \left[ D_2^{(L)} \left( -D_1^{(L)} + 2D_2^{(L)} - D_3^{(L)} \right) + D_2^{(R)} \left( -D_1^{(R)} + 2D_2^{(R)} - D_3^{(R)} \right) \right],
\end{aligned}
\tag{C.1}
$$

$$S_0^{yy}(r) = -\frac{1}{r^2} \cdot \frac{1}{3} \left[ D_3^{(L)} \left( D_1^{(L)} + D_2^{(L)} + D_3^{(L)} \right) + D_3^{(R)} \left( D_1^{(R)} + D_2^{(R)} + D_3^{(R)} \right) \right],$$

$$S_\pi^{yy}(r) = \frac{\ln^{1/2}(r/r_0)}{r} \cdot \frac{1}{3} C_3 (C_1 + C_2 + C_3),$$

$$S_{\pi/3,(1)}^{yy}(r) = \frac{\ln^{1/2}(r/r_0)}{r} \cdot \frac{1}{\sqrt{3}} C_3 (C_2 - C_1),$$

$$S_{\pi/3,(2)}^{yy}(r) = \frac{\ln^{1/2}(r/r_0)}{r} \cdot \frac{1}{3} C_3 (-C_1 - C_2 + 2C_3),$$

$$S_{2\pi/3,(1)}^{yy}(r) = -\frac{1}{r^2} \cdot \frac{1}{\sqrt{3}} \left[ D_3^{(L)} \left( D_1^{(L)} - D_2^{(L)} \right) + D_3^{(R)} \left( D_1^{(R)} - D_2^{(R)} \right) \right],$$

$$S_{2\pi/3,(2)}^{yy}(r) = -\frac{1}{r^2} \cdot \frac{1}{3} \left[ D_3^{(L)} \left( -D_1^{(L)} - D_2^{(L)} + 2D_3^{(L)} \right) + D_3^{(R)} \left( -D_1^{(R)} - D_2^{(R)} + 2D_3^{(R)} \right) \right], \quad \text{(C.2)}$$

$$S_0^{zz}(r) = -\frac{1}{r^2} \cdot \frac{1}{3} \left[ D_1^{(L)} \left( D_1^{(L)} + D_2^{(L)} + D_3^{(L)} \right) + D_1^{(R)} \left( D_1^{(R)} + D_2^{(R)} + D_3^{(R)} \right) \right],$$

$$S_\pi^{zz}(r) = \frac{\ln^{1/2}(r/r_0)}{r} \cdot \frac{1}{3} C_1 (C_1 + C_2 + C_3),$$

$$S_{\pi/3,(1)}^{zz}(r) = \frac{\ln^{1/2}(r/r_0)}{r} \cdot \frac{1}{\sqrt{3}} C_1 (C_3 - C_2),$$

$$S_{\pi/3,(2)}^{zz}(r) = \frac{\ln^{1/2}(r/r_0)}{r} \cdot \frac{1}{3} C_1 (2C_1 - C_2 - C_3),$$

$$S_{2\pi/3,(1)}^{zz}(r) = -\frac{1}{r^2} \cdot \frac{1}{\sqrt{3}} \left[ D_1^{(L)} \left( D_2^{(L)} + D_3^{(L)} \right) - D_1^{(R)} \left( D_2^{(R)} + D_3^{(R)} \right) \right],$$

$$S_{2\pi/3,(2)}^{zz}(r) = -\frac{1}{r^2} \cdot \frac{1}{3} \left[ D_1^{(L)} \left( 2D_1^{(L)} - D_2^{(L)} - D_3^{(L)} \right) + D_1^{(R)} \left( 2D_1^{(R)} - D_2^{(R)} - D_3^{(R)} \right) \right]. \quad \text{(C.3)}$$

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
