# Peer review of "Nonsymmorphic spin-space cubic groups and SU(2)$_1$ conformal invariance in one-dimensional spin-1/2 models"

_SciPost Physics, doi:SciPost Phys. 17, 097 (2024)_

## Round 1 · Referee Report · Anonymous (Referee 1) · 2022-9-15

Strengths

1-This manuscript can be considered as a key reference for anyone interested to learn about the emergence of SU(2) symmetry in discrete Kitaev-like spin chains.

2-The results of the paper are very convincing given a very careful analysis, and given that it combines complementary approaches: group theory, field theory, and numerical density matrix renormalization group.

3-It is very exhaustive, studying symmetry breaking terms, resulting symmetry groups, and resulting effective theories.

4-It is very pedagogical and at the same time delivers new results.

Weaknesses

1-The key ideas of this manuscript have been worked out in multiple previous papers by the authors (Refs.24,34 and many more). But this does not take away from the key role of this paper which is (i) to explore in details general symmetry groups and (ii) provide a full detail exposition.

2-An experimental relevance of the model is not particularly clear.

Report

Previous works have demonstrated an emergent SU(2) symmetry in the Kitaev-Gamma spin chain. The present work presents a breakthrough on these previously-identified research directions, by extending the prediction of emergent SU(2) symmetry to other discrete cubic point groups. In addition to these new results, the paper is particularly well written, containing many useful details, citing relevant references, and hence can also serve as a indispensable reference for a new-comer to the field. I strongly recommend it publication. Few minor comments remarks appear below.

---In the caption of Fig. 1 please add: "This figure is taken from Ref. [34].".
---Below Eq.4 please define $T_a$ (translation by one lattice site) - in addition to the definition of $T_{na}$.
---Below Eq. 23 (or elsewhere) please address the question: is there any numerical evidence for different velocities for the left and right moving sectors?
---Eq.27 typo - second equation should be $\lambda^{-1} \tau - i x$.
---Above Eq.29 "and similar for..."-->"and similarly for...".
---Eq.36 the $\ell$ is probably a typo.
---In the paragraph below Eq.38 "an extend region" -->"an extended region"
---Eqs.41 and 42 and , one of the L subscript should be R.
---Below Eq.103 typo: "are shown are shown"
  • validity: top
  • significance: top
  • originality: top
  • clarity: top
  • formatting: perfect
  • grammar: perfect

Author:  Wang Yang  on 2024-08-03  [id 4676]

(in reply to Report 1 on 2022-09-15)

We thank the referee very much for carefully reading our manuscript and for regarding our paper as “a key reference” in the field. All of the points raised by the referee have been well-taken: The typos have been corrected, and in particular, Sec. 3.5 has been added in the revised manuscript to numerically show that the velocities are indeed different for the left and chiral sectors when the system has neither inversion nor time reversal symmetries.

---

## Round 1 · Referee Report · Anonymous (Referee 2) · 2023-3-1

Strengths

1- important results in a timely problem. 2- Very well written

Weaknesses

none

Report

Emergent symmetries in many body systems are a timely and central theme of modern research. It was already known that a SU(2) symmetry emerges in the so called Kitaev-Gamma spin chain. Here the authors show that SU(2) symmetry emerges as an extended gapless phase in a much more general way in spin 1/2 chain with cubic symmetry in the Hamiltonian. It is shown that all the five nonsymmorphic cubic groups can stabilize SU(2) phases, whereas nonsymmorphic planar groups cannot. The authors also built minimal models for the corresponding nonsymmorphic cubic groups, and provided numerical evidence of emergent SU(2) conformal invariance.This is definitively an important addition to what was know about emergent symmetry in lattice spin models and deserves publication.
  • validity: top
  • significance: top
  • originality: top
  • clarity: top
  • formatting: perfect
  • grammar: perfect

Author:  Wang Yang  on 2024-08-03  [id 4677]

(in reply to Report 2 on 2023-03-01)

We thank the referee very much for regarding our paper as “timely” and “very well written”.

---

## Round 2 · List of Changes

1. The typos pointed out by the first referee have been corrected.

2. Sec. 3.5 is added in the revised manuscript to address the following point raised by the first referee: “Below Eq. 23 (or elsewhere) please address the question: is there any numerical evidence for different velocities for the left and right moving sectors?”.

---

## Editorial Decision

published